# Linking soil biology and chemistry in biological soil crust using isolate exometabolomics

Tami L. Swenson [1], Ulas Karaoz [2], Joel M. Swenson[3], Benjamin P. Bowen[1,4] & Trent R. Northen[1,4]

Metagenomic sequencing provides a window into microbial community structure and metabolic potential; however, linking these data to exogenous metabolites that micro-organisms process and produce (the exometabolome) remains challenging. Previously, we observed strong exometabolite niche partitioning among bacterial isolates from biological soil crust (biocrust). Here we examine native biocrust to determine if these patterns are repro-duced in the environment. Overall, most soil metabolites display the expected relationship (positive or negative correlation) with four dominant bacteria following a wetting event and across biocrust developmental stages. For metabolites that were previously found to be consumed by an isolate, 70% are negatively correlated with the abundance of the isolate's closest matching environmental relative in situ, whereas for released metabolites, 67% were positively correlated. Our results demonstrate that metabolite profiling, shotgun sequencing and exometabolomics may be successfully integrated to functionally link microbial commu-nity structure with environmental chemistry in biocrust.

[1] Environmental Genomics and Systems Biology Division, Lawrence Berkeley National Laboratory, 1 Cyclotron Rd, Berkeley, CA, 94720, USA. [2] Climate and Ecosystems Sciences Division, Lawrence Berkeley National Laboratory, 1 Cyclotron Rd, Berkeley, CA, 94720, USA. [3] Biological Systems and Engineering Division, Lawrence Berkeley National Laboratory, 1 Cyclotron Rd, Berkeley, CA, 94720, USA. [4] DOE Joint Genome Institute, 2800 Mitchell Dr., Walnut Creek, CA, 94598, USA. Correspondence and requests for materials should be addressed to T.R.N. (email: TRNorthen@lbl.gov)

In soils, which harbor the largest terrestrial pool of organic carbon[1], organic matter is largely processed by complex microbial communities. The impact of climate change on these communities and their activities is uncertain[2]. Given the importance of these systems, vast amounts of sequencing data have been and continue to be collected. While metagenomic sequencing provides important insights into community structure and metabolic potential, if unconstrained, such data are often open to multiple interpretations. New approaches are needed to help link the now readily available sequencing data to in situ metabolism in order to better understand the dynamic reciprocity between carbon cycling and microbial community structure.

Soil organic matter (SOM) content and moisture have long been recognized as important factors controlling soil microbial community structure and carbon cycling[3,4]. For example, microbial community diversity and richness are positively correlated with soil organics across diverse ecosystems including polar soils[5], agricultural soils[6] and arid soils[7]. Similarly, soil wetting events are well-known to dramatically alter community structure[8,] including establishing cascades of microbial abundances[9]. Arid lands account for over 40% of Earth's terrestrial surface[10] and are especially sensitive to SOM and moisture content. It is predicted that the aridity of drylands will increase, reducing SOM and microbial community diversity, and that this will impact ecosystem productivity[7,11]. This strong coupling between soil moisture, SOM and community structure is especially important in the arid land topsoil microbial communities known as biological soil crusts (biocrusts), which cover a large fraction of arid regions and are critical in nutrient cycling[12]. Biocrusts exist in a dormant desiccated state and only become metabolically active during infrequent rainfall events[13] and like in other soils, organic matter plays a vital role in retaining moisture and increasing microbial diversity[14].

The mechanisms linking SOM composition and microbial community structure are poorly understood. It is now thought that the organic matter that is cycled by soil microbes is a complex mixture of microbial metabolites[15,16] that can be characterized in detail using soil metabolomics[17,18]. The composition of these exometabolites has a strong impact on community structure, and in turn, these microbes impact the metabolite pool. For example, in some cases, resource competition can reduce microbial diversity through competitive exclusion, whereas cross-feeding can increase microbial diversity. On the other hand, rich sources of SOM may promote microbial diversity through niche divergence[19] and exometabolite niche partitioning[20].

Exometabolomics enables direct examination of how microbes transform the small molecule metabolites within their environment, providing new insights into resource competition and cross-feeding[21]. For this approach, microbes (typically isolates) are cultured in an environmentally-relevant mixture of metabolites and then spent media is profiled to determine the uptake and release of metabolites. Recently, exometabolomics was used to study resource partitioning among sympatric biocrust isolates using complex biocrust-relevant media[20]. This revealed a high degree of substrate specialization, where 13–26% of the detected metabolites were consumed by individual isolates. As microorganisms from diverse taxa continue to be cultivated and examined, this approach holds substantial potential to provide valuable phenotypic information that can link community structure to SOM composition.

Here we exploited the dynamic and tractable characteristics of biocrust to explore the relationships between soil microbes and metabolites for this particular ecosystem after a laboratory-based wetting event. Liquid chromatography-mass spectrometry (LC/MS) soil metabolomics was used to characterize the dynamic metabolite composition of the biocrust soil water and shotgun metagenomic sequencing was used to measure single copy gene markers of the dominant taxa. We then determined the extent to which isolate exometabolite patterns are conserved in situ (within the intact biocrust soil community). While the comparison of a microbe in isolation and in an environmental system is challenging, isolate exometabolomics were performed with media supplemented with lysed cell metabolite extracts to simulate the biocrust environment[20]. Our findings reveal significant agreement between isolate exometabolomics and in situ microbe–metabolite relationships for dominant taxa in biocrust. To the best of our knowledge, this is the first study using isolate exometabolomics to link microbial community structure to soil chemistry.

## Results

**Experimental setup.** Correlation analyses between microbes and metabolites were facilitated by monitoring biocrust samples across two variables: laboratory-based wetting and ecological succession (Fig. 1a). Four successional stages of biocrust were used, ranging from early/ young (labeled as 'level A') to late/ mature (labeled as 'level D') (Supplementary Fig. 1). We then compared our current results with previous laboratory-derived knowledge of substrate preferences for four dominant microorganisms by relating the abundance of these bacteria to soil metabolites measured in the intact biocrust system (Fig. 1b). In situ, the general assumption is that as a particular microbe grows and increases in abundance in a community, consumed metabolites will decrease and display a negative-correlation relationship. Conversely, metabolites that are known to be released by a microbe are predicted to concurrently increase and display a positive-correlation relationship with growth (Fig. 1b).

**Metabolite and microbe dynamics.** The metabolic activity caused by wetting was monitored at various time points ranging from immediate (3 min) to longer-term (49.5 h) across four biocrust successional stages. Biocrust soil water was analyzed by LC/MS, resulting in the identification of 85 metabolites for the experimental data set using authentic chemical standards (Fig. 2 and Supplementary Data 1). All 85 metabolites identified in the active biocrust samples changed at least two-fold (between minimum and maximum peak areas) across both wetting and successional stages (Fig. 2). Wetting duration had a stronger impact on metabolite dynamics vs. successional stage (Supplementary Fig. 2). Hierarchical clustering of metabolite patterns revealed three distinct clusters (Fig. 2). The first cluster (cluster 1, Fig. 2) consisted of most (5 out of 7) of the detected fatty acids (palmitate, myristate, stearate, laurate, decanoate), which were most abundant at the first time point (3 min) for all successional stages, and gradually decreased with time. The largest cluster (cluster 2, Fig. 2) was enriched with the majority of amino acids and nucleobases, which peaked in abundance at the early to early-mid time points. Within this cluster, the earliest metabolites included polar amino acids (glutamine, glutamate, asparagine, 4-oxoproline, aspartate and lysine) and the nucleobases uridine, guanosine and cytidine. The final cluster (cluster 3 in Fig. 2) contained metabolites most abundant at late time points and in more mature biocrust (e.g., salicylate, panthothenate, nicotinate, xanthine, creatinine). These dynamics, especially metabolite consumption, were largely of biological origin as demonstrated by comparison with the killed controls. Thirteen metabolites that were detected in the active samples were not detected in the killed controls presumably due to the lack of biologically activity. Of the 72 metabolites that were detected in the killed controls, all but 19 were significantly different ($p < 0.05$, by two-way ANOVA and Tukey's post hoc test) from the active samples and qualitatively

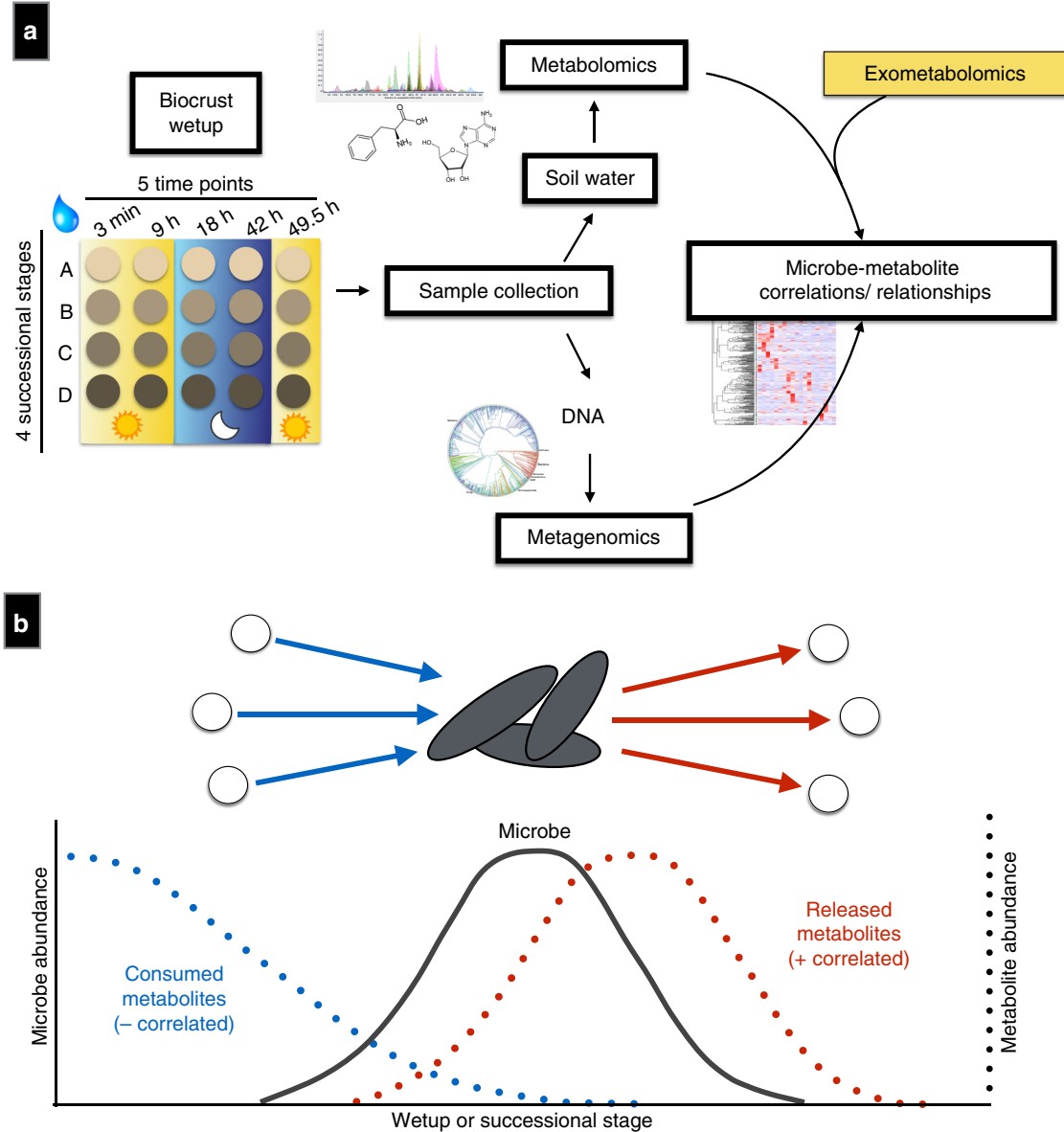

**Fig. 1** Experimental workflow and biocrust microbe–metabolite relationship predictions. **a** Biocrust wetup metabolomics and metagenomics experimental setup and analysis. To study microbe–metabolite relationships in situ, biocrusts from four successional stages were wetup and sampled at five time points (total $n = 100$). Biocrust soil water was removed and analyzed by liquid chromatography/ mass spectrometry ($n = 5$ for each group) and biocrust DNA was extracted for shotgun sequencing ($n = 1$ for each group). Metagenome-estimated genome and metabolite abundances were analyzed through Spearman rank correlations to determine microbe–metabolite relationships and compared to the expected relationships based on isolate exometabolomic studies. **b** Exometabolomics-based in situ microbe–metabolite relationship prediction. The hypothesis is that isolate exometabolomics can be used to predict microbe–metabolite patterns in situ based on microbial abundance: Across wetting and successional stages, microbes change in abundance and negatively correlate with metabolites that they consume and positively correlate with metabolites that they release (metabolites are indicated by dotted lines)

all show different dynamics (Supplementary Data 2 and 3; Supplementary Fig. 3).

Biocrust microbial community structure was inferred by shotgun metagenomics using a genome-centric pipeline[22]. In several recent reports[23–26], ribosomal protein genes were used as phylogenetic markers from shotgun sequencing data (rather than the more classical 16s ribosomal RNA gene) because they exist as single copies in almost all genomes, assemble well from the metagenome data sets (typically better than 16s), are well-conserved and have produced higher resolution phylogenetic trees[25]. We identified a set of 17 previously-benchmarked single copy universal ribosomal protein genes[27] in our biocrust data set and for community analysis, we selected rplO (ribosomal protein

L15), which had the most extensive community coverage based on the total number of assembled genes across our data set (Supplementary Table 1).

On the basis of rplO genes, 466 distinct biocrust genomes were identified across all conditions (Supplementary Data 4). As observed for biocrust metabolites, community structure was primarily driven by wetting duration. At the phylum level, the most drastic change was a shift from a Cyanobacteria-dominated community at early time points (17–28% at 3 min to 1–3% by 49.5 h) to a Firmicutes-dominated community by 49.5 h (4–5% at 3 min to 19–39% by 49.5 h) (Supplementary Fig. 4). Other dominant phyla included Proteobacteria and Actinobacteria, which appeared to be indifferent to wetting (i.e., their relative

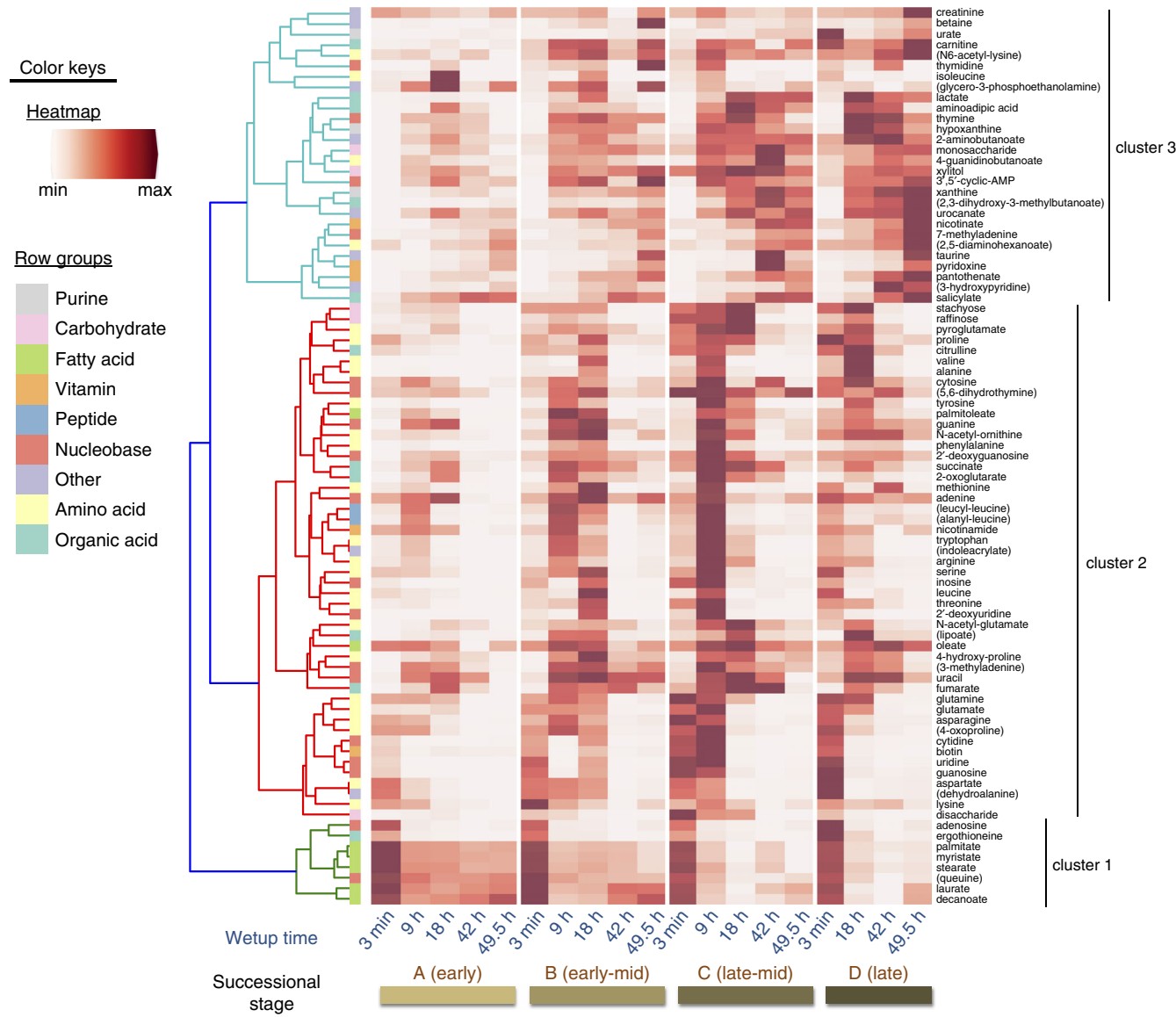

**Fig. 2** Metabolite patterns detected in biocrust soil water. Metabolite dynamics (85 metabolites displayed as the average peak area, normalized across each row) were observed in biocrust soil water across wetting and successional stages. Unique patterns are indicated by cluster 1 (early metabolites including fatty acids), cluster 2 (early-to-mid time point metabolites) and cluster 3 (late metabolites). Putative metabolites are indicated by parentheses. *n* = 2–5 for each group

abundance was more evenly-distributed across wetting) (Supplementary Fig. 4).

In order to use previous culture-based isolate exometabolomic analyses to examine the link between microbe and metabolite dynamics in intact biocrust, we conducted an unbiased *rplO* gene sequence comparison between *rplO* genes assembled from the native biocrust and previously profiled biocrust bacterial isolates[20] (obtained from the same field site). Fortunately, four isolates matched relatively abundant native biocrust bacteria at the species or genus level (Table 1). Further comparative analyses were conducted to calculate genome average nucleotide identity (ANI) between isolate genomes and their related metagenome-assembled genomes (MAGs) to which the *rplO* gene was co-binned. This validated that each isolate and their closest matching environmental relative were of the same genus or species, following the conventions described in a previous report[28] (Table 1). Additional validation was obtained by comparing gene average amino acid identity (AAI) (Supplementary Table 2) and

other ribosomal protein genes (Supplementary Data 5) from within each MAG to the corresponding isolate. On the basis of these results, these four isolate-related environmental bacteria were selected for exometabolomics comparisons: *Microcoleus* sp. (a filamentous Cyanobacterium and primary producer), two Firmicutes (referred to here as *Bacillus* sp. 1 and *Bacillus* sp. 2) and *Blastococcus* sp. (an Actinobacterium) (Table 1, Supplementary Fig. 5).

Remarkably, together, these four bacteria accounted for ~30% of the entire microbial community in many of the biocrust samples (Supplementary Fig. 6). The overall most abundant microorganism, *Microcoleus* sp. is known to be a pioneer species responsible for initial soil stabilization and biocrust formation[29] and in our study was the most dominant in early wetup biocrust, accounting for 10–25% of the entire microbial community at 3 min across all successional stages (Supplementary Fig. 6). The two Firmicutes, *Bacillus* sp. 1 and *Bacillus* sp. 2, are likely to be physically-associated with *Microcoleus* filaments[20] and their

**Table 1 Exometabolite-profiled isolates and their closest matching relative in biocrust**

| Isolate ID from Baran et al.[20] | Closest matching biocrust relative | Taxonomy | rplO[a] (%) | ANI (%) |
|---|---|---|---|---|
| *M. vaginatus* PCC9802 | *Microcoleus* sp. (*rplO* 1) | Cyanobacteria (p)/ Oscillatoriophycideae (c)/ Oscillatoriales (o)/ Microcoleaceae (f)/ Microcoleus (g) | 92.0 | 94.4 |
| D1B51 | *Bacillus* sp. 1 (*rplO* 2) | Firmicutes (p)/ Bacilli (c)/ Bacillales (o)/ Bacillaceae (f)/ Bacillus (g) | 86.3 | 75.4 |
| L2B47 | *Bacillus* sp. 2 (*rplO* 60) | Firmicutes (p)/ Bacilli (c)/ Bacillales (o)/ Bacillaceae (f)/ Bacillus (g) | 87.6 | 76.6 |
| L1B44 | *Blastococcus* sp. (*rplO* 7) | Actinobacteria (p)/ Actinobacteria (c)/ Geodermatophilales (o)/ Geodermatophilaceae (f)/ Blastococcus (g) | 87.1 | 72.9 |

a Similarity in *rplO* sequence (isolate vs. biocrust microorganism)

relative abundance increased during wetting. The most abundant of these, *Bacillus* sp. 1, was a mid-wetup responder and peaked at 9 h for successional levels A, B and D (16–24% of the community) and at 18 h for successional level C (24% of the community) (Supplementary Fig. 6). *Bacillus* sp. 2 reached its peak abundance at later time points, accounting for up to 3% of the community by 42 h in successional level C, noting that by the later time points the community was less dominated by any one particular microorganism (Supplementary Fig. 6). Finally, *Blastococcus* sp. abundance was found to be relatively resistant to wetting and was somewhat evenly distributed across all conditions (0.1–2% of the community) (Supplementary Fig. 6).

**Microbe–metabolite relationships in situ vs. in culture**. To determine how isolate substrate preferences impact in situ exometabolite composition, we evaluated microbe–metabolite relationships, focusing on metabolites known to be released or consumed by biocrust isolates and their closest matching environmental relatives, *Microcoleus* sp., *Bacillus* sp. 1, *Bacillus* sp. 2 and *Blastococcus* sp. The general expectation was that released metabolites would be positively correlated with the relative bacterial abundance while consumed metabolites would be negatively correlated (Fig. 1b) across environmental variables, including wetting and successional stage. To link previous isolate exometabolomics data with the current biocrust exometabolome data set (Fig. 2), we determined the direction and degree of correlation between the metabolites that were previously found to be consumed and released by biocrust isolates[20] and closest matching environmental relatives (*Microcoleus* sp., *Bacillus* sp. 1, *Bacillus* sp. 2 and *Blastococcus* sp).

Of the 85 metabolites identified in the biocrust soil water, 32 matched the isolate exometabolome data set. Nine of these displayed temporal patterns that were not significantly different than the killed control and were excluded from this analysis since abiotic controls on these metabolites could not be ruled out (Supplementary Data 6). Spearman's rank correlation was used to assess the directionality (positive vs. negative correlations) of biocrust microbe–metabolite relationships. Strikingly, of the 48 microbe–metabolite relationships evaluated (Supplementary Fig. 7), 69% had the expected directionality that would be predicted based on isolate exometabolomics data. This overall observation of correct directionalities is significantly higher than what would be expected by chance ($p$-value = 0.01 by two-tailed binomial test; Supplementary Data 7).

We next combined our microbe-metabolite correlation data with the isolate exometabolomics data in order to visualize a wetting-induced dynamic exometabolomic foodweb that may result from the release of metabolites by a primary producer (e.g., *Microcoleus* sp.) and consumption of metabolites by heterotrophs (e.g., *Bacillus* sp. 1 and 2) (Fig. 3). For simplicity, *Blastococcus* sp. is not shown in Fig. 3 since this microorganism did not display sequential responses to wetting. It should be noted that the

bacteria specified here likely represent metabolically similar groups of organisms that release or consume the same metabolites. Of the set of metabolites that were most highly-released by *M. vaginatus* PCC 9802[20], 14 of these were detected in the biocrust soil water and half were positively correlated with *Microcoleus* sp. across wetting and successional stages (Fig. 3 and Supplementary Fig. 7). While *Microcoleus* sp. was most abundant immediately following wetting, most of these metabolites (71%) reached their highest level during the first three time points (3 min, 9 h or 18 h; Supplementary Fig. 7) just after the *Microcoleus* sp. spike, suggesting release by *Microcoleus* sp. followed by consumption by heterotrophs as they become metabolically active.

Consistent with a heterotrophic lifestyle, most metabolites evaluated displayed a negative relationship with the relative abundances of *Bacillus* sp. 1, *Bacillus* sp. 2 and *Blastococcus* sp. Of the metabolites that were consumed by the *Bacillus* sp. 1. related isolate, D1B51 (Table 1)[20], six were detected in the current biocrust soil water samples and reached their highest level early-on (at either 3 min, 9 or 18 h), decreasing just after the *Bacillus* sp. 1 peak (Supplementary Fig. 7). Four of these metabolites were negatively correlated with *Bacillus* sp. 1, consistent with metabolite consumption, and all three D1B51-released metabolites were positively correlated with *Bacillus* sp. 1 (Fig. 3 and Supplementary Fig. 7). As for the less dominant microorganisms, the late-wetup responder, *Bacillus* sp. 2, was negatively correlated with all seven metabolites that were consumed by the related isolate (L2B47) and positively correlated with all four isolate-released metabolites (Fig. 3 and Supplementary Fig. 7). Furthermore, *Blastococcus* sp., was negatively correlated with eight out of the ten metabolites that were consumed by the related isolate (L1B44). Finally, the closest matching environmental relatives to the three remaining exometabolomic-profiled isolates (L1B56, D1B2, and D1B45) accounted for 0.1% or less of the microbial community in our metagenomes and thus, not surprisingly, did not display exometabolite-based microbe–metabolite relationships (data not shown).

**Transcriptomics support soil microbe–metabolite relationships**. Transcriptomics has the potential to test if gene expression is consistent with predicted substrate utilization and release patterns. As an initial proof of concept, we further analyzed the data obtained from a previous study that evaluated *M. vaginatus* gene expression following wetup and drydown in biocrusts collected from the same field site[30]. We found that pathways involved in the biosynthesis of amino acids (KEGG pathways 'biosynthesis of amino acids', 'phenylalanine, tyrosine and tryptophan biosynthesis' and 'valine, leucine and isoleucine biosynthesis') all increased dramatically during early wet-up (Supplementary Fig. 8, Supplementary Data 8). In contrast, pathways involved in the degradation of these same metabolites were relatively constant ('phenylalanine metabolism') or only slightly increased

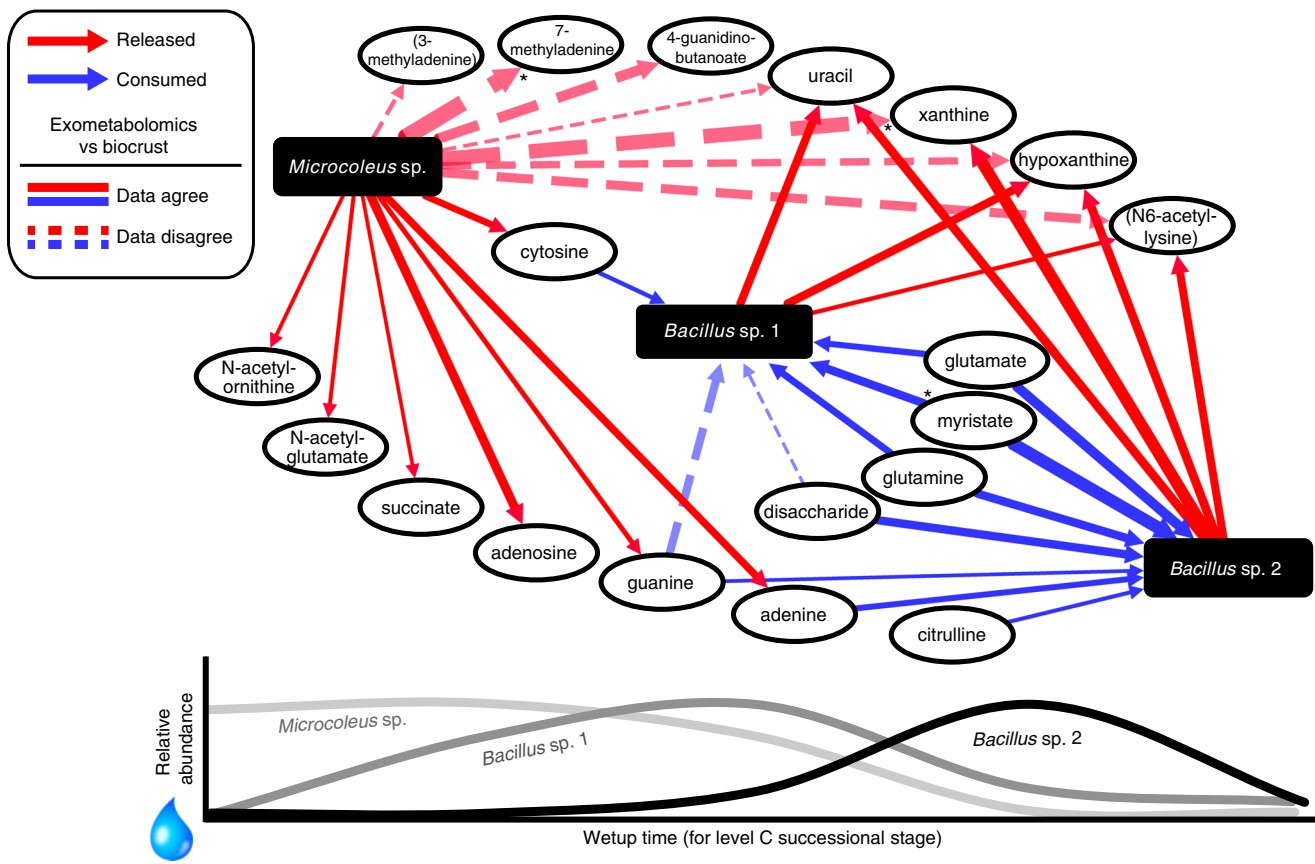

**Fig. 3** Simplified biocrust foodweb for three dominant biocrust bacteria based on combining isolate exometabolomics data with in situ microbe–metabolite relationships. This network displays the relationships between metabolites and three dominant bacteria (or metabolically similar organisms) as they increase and decrease in relative abundance across wetting and successional stages in biocrust. The lower line plot corresponds to real relative abundance measurements for the three bacteria in level C successional stage biocrust. As *Microcoleus* sp. increases in relative abundance immediately after wetting, many metabolites released by the closest-related isolate are positively correlated with *Microcoleus* sp. in biocrust (solid red arrows) and as the two *Bacilli* increase in relative abundance (first *Bacillus* sp. 1 then *Bacillus* sp. 2), most metabolites consumed by the closest-related isolate decrease and are negatively correlated with these bacteria in biocrust (solid blue arrows) and most released metabolites are positively correlated (solid red arrows). Dotted arrows indicate metabolites that are released (red) or consumed (blue) by isolates, but did not display the expected relationship with that microorganism in situ. The thickness of the line corresponds to the absolute value of the Spearman's *rho* correlation coefficient. The overall expected directionality (solid lines vs. dotted lines) was significant as determined by the exact binomial test (two-tailed *p*-value = 0.01). *FDR-adjusted $p < 0.05$ for individual microbe-metabolite Spearman correlations

('tryptophan metabolism' and 'valine, leucine and isoleucine biosynthesis') following wet-up (Supplementary Fig. 8). These observations are consistent with the release of these metabolites by *M. vaginatus* PCC 9802[20] and the observed immediate increase of most amino acids in biocrust soil water in the present study.

## Discussion

Sequencing has the potential to link exometabolite composition to specific microbes based on genome annotations. However, with these data alone, relating metabolic potential to activity is challenging. Despite this challenge, sequencing and other approaches have started to shed light on the impact individual microorganisms[31], microbial genes[32] and enzymatic activities[33] have on the chemistry within their environment. Here we evaluated exometabolite profiles of individual bacteria in order to link soil exometabolites to bacteria in biocrust, a critical ecosystem that lends itself to studies of community responses to soil wetting.

We found that a biocrust wetting event set in motion an immediate cascade of microbial activities marked by a drastic shift in community structure. The dominance of Cyanobacteria during early time points is consistent with previous reports[34] as is the subsequent Firmicutes-bloom[9,35]. The Firmicutes phylum consists mostly of gram-positive, spore-forming bacteria with rapid generation times, enabling them to 'bloom' upon soil wetting[36,37]. The observed switch from a Cyanobacteria-dominated community to a Firmicutes-dominated community (mostly *Bacillus* sp. 1 in this study) agrees with our observations of metabolite release by the dominant photoautotroph (*Microcoleus* sp.)[20] followed by consumption and growth of diverse heterotrophs (e.g., Firmicutes), possibly including symbiotic nitrogen-fixers[38]. While we did not observe evidence of fixed nitrogen transfer into Cyanobacteria, this process may occur during dry-down, when nitrogen-rich nutrients may be released upon the mother cell lysis stage of sporulation[39].

It has been suggested that copiotrophic microorganisms (e.g., many Firmicutes) are superior competitors for a limited number of compounds, whereas oligotrophs (e.g., many Actinobacteria) support a more stable population by using a wider range of substrates[40]. Our previous isolate exometabolomics work is consistent with this view by showing that the two Firmicutes isolates depleted the narrowest range of substrates (10%), whereas the two Actinobacteria used almost twice as many[20]. During

biocrust wetting, we found that, unlike the boom-bust cycle of Firmicutes, the Actinobacteria phylum (such as *Blastococcus* sp.) appeared more resistant to wetting duration. This provides limited evidence that utilization of diverse substrates, which is consistent with oligotrophy, may enable slow but continuous growth under conditions with highly dynamic exometabolite pools.

The wetting-induced initiation of microbial community dynamics coincided with an immediate release of metabolites into biocrust soil water. Killed controls were used to differentiate biotic processes of interest from abiotic dynamics driven by metabolite leakage, slow diffusion, mineral sorption, thermolysis and photolysis. As expected, the active biocrusts exhibited extremely different dynamics, with dramatic, near complete depletion of many metabolites, whereas the killed controls (with the exception of the antioxidant ergothioneine) show a gradual increase in most metabolites presumably as they are released from the dead biomass (Supplementary Fig. 3). Some likely biological mechanisms involved in the active metabolite "pulse" include osmotic stress induced release of solutes (e.g., proline, betaine and disaccharides) to maintain cell integrity[41,42] as well as photosynthate release from the primary producer. These metabolite dynamics that followed wetting resulted in strong microbe–metabolite relationships that were tracked across time and interestingly, most of these strong relationships (correlations) were conserved from one successional stage to another for the four bacteria that were examined (*Microcoleus* sp., *Bacillus* sp. 1 and 2 and *Blastococcus* sp.) (Supplementary Fig. 9). This supports the notion that the content of water-soluble SOM in these biocrusts, to a large degree, originates from and is controlled by microbes[15] and the composition of this pool may be predictable if a change in microbial community structure is anticipated. This finding has particular significance for biocrusts, since changes in temperature and rainfall are expected to shift microbial community structure[43,44]. As a result, these alterations are expected to impact SOM cycling, especially if there is loss of taxa responsible for utilization or production of specific SOM components.

Next, we explored the connection between the observed microbe–metabolite relationships in biocrust and culture-based exometabolite profiles, focusing on metabolites found to be controlled by biological mechanisms. Comparison of the two data sets was facilitated by the fact that the isolates were obtained from the same field site. Therefore, we used the phylogenetic marker, *rplO* (which had the most coverage in our data set) to relate exometabolite-profiled isolates to native biocrust bacteria. Isolate genomes and their closest matching environmental MAGs had between 73–94% genome ANI values and were 86–92% identical in their *rplO* sequences (Table 1). These relationships were further confirmed by comparing other ribosomal protein genes, which all ranged between 72–100% identical between each isolate and its environmental MAG (Supplementary Data 5). While there is a general lack of consensus of valid isolate-to-native population comparisons, our values indicate matching at approximately the genus or species level[28]. Much of the difficulty in these comparisons is due to genomic heterogeneity within environmental samples[45]. However, despite these complexities with determining exact phylogenetic distances, we observed functional similarity between the isolate and environmental exometabolomics data sets, which is consistent with the view that many metabolic traits are conserved at the genus level[46]. An exciting alternative explanation for our observations is that the clades that these isolates are members of exhibit the same cohesive dynamics as the closest-matching bacteria in the biocrust[47].

Overall, we found that isolate exometabolomic patterns were conserved in the intact biocrust soil microbial community. The expected directionality (positive or negative microbe–metabolite relationships) (solid arrows in Fig. 3) was significantly higher

than predicted by chance, indicating a linkage between laboratory observations and in situ soil activities. While most metabolites displayed the expected patterns, some biocrust soil water metabolites (e.g., uracil, N6-acetyl-lysine, hypoxanthine and xanthine) behaved inconsistently with *M. vaginatus* PCC 9802 exometabolite profiles. However, these were also released by and positively correlated with *Bacillus* sp. 1 and 2. Deconvoluting this may be possible using dynamic utilization models[48,49] to account for the relative contributions of the two microorganisms. Ultimately, this same approach could be used to account for rare community members that may also have an impact on the exometabolite pool or may alter the metabolism of other microbes[50,51]. Although outside of the scope of the current work, we anticipate that these substrate-genome linkages could be further tested and refined by using other approaches. Stable isotope probing coupled with labeled DNA sequencing[38,52] and integrated NanoSIMS and FISH imaging[53,54] may be used to examine the spatial localization of microbes and their activities.

We next used the biocrust microbe–metabolite relationships to display a simplified dynamic exometabolomics web describing the wetting response of three dominant bacteria in biocrust (Fig. 3). While this network portrays the dynamic substrate preferences of three specific native biocrust bacteria, there are many functionally-similar microorganisms that could fall into the categories of 'early responders', 'mid-responders' and 'late-responders' (grouping with *Microcoleus* sp. and the two *Bacilli* sp., respectively) for a wetting event. However, based on our available data and analyses, this network displays the role of *Microcoleus* sp. as the biocrust primary producer, releasing many metabolites, which stimulates the growth and metabolite consumption by heterotrophs (in this case, the two heterotrophic Firmicutes). Our findings suggest unique microorganismal roles in the biocrust foodweb, including the release of nucleobases (uracil, hypoxanthine and xanthine) by the two Firmicutes, which is consistent with our earlier reports of heterotrophs releasing these compounds[55]. This may reflect a nitrogen-scavenging mechanism by consuming N-containing substrates (cytosine, adenine, guanine, and histidine), producing uracil, hypoxanthine, and xanthine as byproducts, which may then be consumed by late-responding microorganisms. Even though these observed functional linkages are for a fraction of the biocrust microbial community, these small yet significant pieces of the puzzle have the potential to help understand and predict nutrient cycling in terrestrial microbial ecosystems[56] analogously to the many microorganisms that have been linked to specific transformations within marine ecosystems. For example, Cyanobacteria release and reuptake organic carbon[57], a variety of uncultured taxa utilize dissolved proteins[52] and SAR11 bacteria assimilate amino acids and dimethylsulfoniopropionate[58].

We attribute much of the success of this study to the suitability of the relatively simplified biocrust soil ecosystem. One such advantage is that the biocrust used in this study is primarily quartz sand, facilitating metabolite analysis compared to many other soils which are typically rich in clays and other strongly-sorptive mineral surfaces[59]. Accurately representing the competition between microbes and mineral surfaces would require additional studies examining mineral-metabolite sorption dynamics[60,61]. Another simplifying factor is that the biocrust community, unlike many other soils, is dominated by a few bacteria, greatly enabling accurate correlations between taxa and metabolites. We anticipate that in order to more accurately predict microbe–metabolite relationships for more diverse communities and complex environments, a large number of relevant taxa would need to be subjected to exometabolite profiling. Additionally, accounting for switching between metabolic states will require profiling under diverse environmental conditions. For

example, the discrepancy between metabolites that were released by *M. vaginatus* PCC 9802[20], but were not correlated with *Microcoleus* sp. abundance in biocrust may be due to different metabolic processes occurring during the day (photosynthesis) vs. night (respiration) (i.e., Diel cycle)[30,62]. Thus, modeling approaches will be required to account for metabolic state switching among other processes. One exciting possibility of expanded exometabolomic data sets, is that knowledge of uptake and release of metabolites can be used as boundary constraints for flux-balance analysis in metabolic models[63] and to inform trait-based models[64,] providing a genome-scale approach for linking soil metabolites with metagenomic data. For example, OptCom[65], a multi-level and multi-objective flux balance analysis framework to understand metabolism within microbial communities, which currently primarily relies on genomic information, could be used in conjunction with exometabolomic data.

In conclusion, this study suggests that, for biocrust, many genotype-to-phenotype relationships are conserved from test tube to soil. These conserved isolate exometabolite patterns begin to provide a functional link between in situ community structure and metabolite composition. We expect that exometabolomic characterization of additional taxa and determination of mineral-metabolite sorption dynamics, under a range of environmentally relevant conditions (e.g., day/night cycles), integrated with modeling approaches will further enhance the predictive power of these relationships. These studies may help pave the way for interpretation and use of metagenomic and metatranscriptomic approaches for linking soil chemistry to soil microbiomes to define exometabolite webs of microbes in complex ecosystems.

## Methods

**Materials**. LC/MS-grade water and LC/MS-grade methanol (CAS 67-56-1) were from Honeywell Burdick & Jackson (Morristown, NJ). LC/MS-grade acetonitrile (CAS 75-05-8) and ammonium acetate (CAS 631-61-8) were from Sigma-Aldrich (St. Louis, MO). LC/MS internal standards included MOPS (CAS 1132-61-2), HEPES (CAS 7365-45-9), 3,6-dihydroxy-4-methylpyridazine (CAS 5754-18-7), 4-(3,3-dimethyl-ureido)benzoic acid (CAS 91880-51-2), $d_5$-benzoic acid (CAS 1079-02-3) and 9-anthracenecarboxylic acid (CAS 723-62-6) from Sigma-Aldrich.

**Sample collection**. Petri dishes ($6 \text{ cm}^2 \times 1$ cm depth) were used to core biocrust samples from the Green Butte Site near Canyonlands National Park (38°42'54.1" N, 109°41'27.0" W, Moab, UT, USA). Samples were collected along an apparent maturity gradient of Cyanobacteria-dominated biocrusts ranging from light, young (level A) to darker, more mature (level D) (Supplementary Fig. 1). Dry samples collected in the field were maintained in a dark desiccation chamber in the laboratory for approximately 11 months until lab-based experiments were performed consistent with previous reports[66]. Biocrusts are adapted to prolonged desiccation though it is important to note that the duration of desiccation likely affects metabolite composition and biological responses.

**Biocrust wetting**. Biocrust (0.5 g) was transferred to each well within 12-well plates. Sterile LC/MS-grade water (1 mL) was added to each sample and placed under a 12 h light (~300 μmol photons/m²s)/ 12 h dark cycle. Microcosms were completely enclosed by aluminum foil to prevent infiltration by outside light sources. At each time point (3 min, 9 h, 18 h, 42 h and 49.5 h), biocrust and soil water were removed and placed in 2 mL Eppendorf tubes and 500 μL of additional water was used to rinse out the wells and added to the sample. Tubes were centrifuged at $5000 \times g$ for 5 min and supernatant (biocrust soil water) was pipetted and placed in new 2 mL tubes. Remaining biocrust was stored at −80 °C until nucleic acid extraction was performed. There were five biological replicates, five time points and four successional stages of biocrust resulting in 100 total samples. An extraction control (three replicates of water, no soil) was included and sampled at 49.5 h to evaluate background contamination. To control for abiotic metabolite processes (such as sorption, photodegradation, etc), a separate experiment was performed where soil water was sampled along the same time points (5 sampling time points in triplicate) from killed (autoclaved four times) biocrust from a late successional stage.

**Metabolite extraction and LC/MS analysis**. Biocrust soil water samples (1.5 mL) were lyophilized and resuspended in methanol (200 μL) containing internal standards (2–10 μg/mL) and filtered through 96-well Millipore filter plates (0.2 μm PVDF) by centrifuging at $1500 \times g$ for 2 min. Samples were analyzed using normal-phase LC/MS with a ZIC-pHILIC column ($150 \times 2.1$ mm, 3.5 μm 200 Å, Merck Sequant, Darmstadt, Germany) using an Agilent 1290 series UHPLC (Agilent Technologies, Santa Clara, California, USA). Chromatographic separation was achieved using two mobile phases, 5 mM ammonium acetate in water (A) and 90% acetonitrile with 5 mM ammonium acetate (B) at a flow rate of 0.25 mL/min with the following gradient: 100% B for 1.5 min, decreased to 50% B by 25 min, held until 29.9 min then returned to initial conditions by 30 min with a total runtime of 40 min. Column temperature was maintained at 40 °C. For MS, negative mode data were acquired on an Agilent 6550 quadrupole time-of-flight mass spectrometer and positive mode data were acquired on a Thermo QExactive (Thermo Fisher Scientific, Waltham, MA). Fragmentation spectra (MS/MS) were acquired for some metabolites using collision energies of 10–40 eV. For the killed biocrust controls, samples were analyzed using an Agilent 1290 series UHPLC with a SeQuant ZIC-HILIC column ($150 \times 2.1$ mm, 5 μm, 200 Å, MilliporeSigma, Billerica, MA). The two mobile phases used were 5 mM ammonium acetate in water (A) and 95% acetonitrile with 5 mM ammonium acetate (B) at a flow rate of 0.45 mL/min with the following gradient: 100% B for 1.5 min, decreased to 65% B by 15 min then 0% B by 18 min, held until 23 min then returned to initial conditions by 25 min with a total runtime of 30 min. The negative and positive MS data for the killed controls were obtained using a Thermo QExactive.

**Metabolite identification and statistical analysis**. Metabolomics data were analyzed using Metabolite Atlas[67] (https://github.com/biorack/metatlas) in conjunction with the Python programming language to construct extracted ion chromatograms corresponding to metabolites previously detected in biocrust or contained within our in-house standards library. Note that this same analysis could be done with other open-source software packages. Authentic standards were then used to validate assignments based on two orthogonal data (accurate mass less than 15 p.p.m., retention time within 1 min and/or MS/MS matching major fragments) relative to standards and/or the Metlin database[68,69] and are provided in Supplementary Data 1. Metabolites that did not match orthogonal measures were classified as putative and are indicated by parentheses in figures. Internal standards were assessed from each sample to ensure peak area and retention times were consistent from sample-to-sample. Quality control mixtures were included at the beginning, end and throughout the runs to ensure proper instrument performance (*m/z* accuracy and retention time and peak area stability). Sample QC failed (internal standards were not present within the specified retention time window) for some replicates including all 9 h wetup level D samples and were not included for further analyses.

To explore the degree of variation in biocrust metabolite profiles across wetting and successional stages, biocrust samples were PCA-ordinated based on their metabolite profiles. Significance between temporal patterns from active biocrust and killed control samples was analyzed using the anovan and multcompare functions in Matlab R2016A with an alpha of 0.05 corresponding to 95% confidence level and Tukey's honestly significant difference test where time is considered a continuous variable and succession is considered a categorical variable (Supplementary Data 3).

**DNA extraction, sequencing and microbial annotation**. DNA was extracted from biocrust (0.25 g) using the MoBio Powersoil DNA isolation kit (MoBio Laboratories, Inc, Carlsbad, CA) resulting in 100 μL of eluted DNA. Library preparation and sequencing were done at the QB3 facility at the University of California, Berkeley using Illumina HiSeq4000 (see supplementary methods for details on metagenome analysis). In recent studies[24,25,70], ribosomal protein genes have been used as phylogenetic markers as an alternative to the more classical 16s ribosomal RNA gene. Ribosomal protein genes exist as single copies in almost all genomes, assemble well from metagenome datasets, are well-conserved and have produced higher resolution phylogenetic trees[25]. Given these advantages, the 50S ribosomal protein L15 (*rplO*) gene had the most extensive community coverage for our data set and was therefore used as a phylogenetic marker to examine the relative abundance of individual microorganisms within the microbial community across wetting and successional stages. The *rplO* genes from the genomes of the seven exometabolite profiled biocrust bacterial isolates[20] were compared to all the *rplO* genes recovered from biocrust MAGs. Those with the highest percent similarity were considered the "closest matching environmental relatives" to the isolates and are reported in Table 1. Average nucleotide and amino acid identity metrics were calculated using the 'enveomics' tool[71].

**Microbe-metabolite correlations**. Correlations were used to identify microbe–metabolite relationships across both wetting and successional stages. Spearman's rank (*rho*) correlation coefficients for every pairwise (microbe–metabolite) relationship and *p* values (unadjusted and FDR-adjusted) were calculated using the cor() stats function in R. A Spearman's *rho* value greater than or equal to 0.5 was considered "highly correlated" and less than or equal to −0.5 was considered "highly negatively correlated". To test if the overall observed directionality (positive vs. negative correlations) was due to chance rather than as would be predicted based on exometabolomics (release vs. consumption), the exact binomial test was conducted using R (binom.test) with a total of 48 "trials" or observed microbe-metabolite interactions (Supplementary Data 7).

**Exometabolomics comparison and analysis**. Of the 85 metabolites detected in biocrust soil water, 32 were previously analyzed for consumption and release by biocrust isolates[20]. Of those metabolites, 23 were selected for further analyses since they were considered to be biologically-controlled (they displayed temporal patterns that were significantly different from the killed controls). For continued analyses of the isolate exometabolomics data here, fold-change was calculated by dividing the average peak area of each metabolite in (isolate) inoculated spent media by the non-inoculated control spent media (raw data can be found in the Supplemental Table in Baran et al.[20]). A metabolite was considered consumed if the fold-change was 0.5 or less and released metabolites had a fold-change of 2 or greater.

**Microcoleus gene expression analysis**. *Microcoleus* genes from Supplementary Table 6 in Rajeev et al.[30] were categorized into KEGG pathways (for genes containing KEGG ID numbers), which are summarized in Supplementary Data 8. Analyses focused on pathways that are primarily anabolic or catabolic for metabolites that were released by *Microcoleus* PCC 9802. The average fold-change (relative to dry biocrusts) and standard errors were calculated for all genes belonging to pathways of interest.

**Data availability**. Sequencing data that support the findings of this study are available at the NCBI Sequence Read Archive (project accession number PRJNA395099; sample accession numbers SRX3024638 to SRX3024657, https://www.ncbi.nlm.nih.gov/bioproject/395099) and metabolomics data have been deposited in the EMBL-EBI MetaboLights database[72] with the identifier MTBLS492 (http://www.ebi.ac.uk/metabolights/MTBLS492). Other relevant data supporting the findings of the study are available in this article and its Supplementary Information files, or from the corresponding author upon request.

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

## Acknowledgements

This work was funded by the Office of Science Early Career Research Program, Office of Biological and Environmental Research, of the U. S. Department of Energy under contract number DE-AC02-05CH11231 to Lawrence Berkeley National Laboratory. DNA was sequenced using the Vincent J. Coates Genomics Sequencing Laboratory at UC Berkeley, supported by NIH S10 OD018174 Instrumentation Grant. We thank Rebecca Lau for technical assistance in biocrust sample collection and experimentation.

## Author contributions

T.L.S. and T.R.N. conceived the study, designed the experiments and wrote the manuscript. T.L.S. performed the experiments. T.L.S. and B.P.B. analyzed the metabolomics data. U.K. analyzed the metagenomics data. T.L.S. and J.M.S. conducted correlation and statistical analyses. All co-authors commented on the design of experiments, data analysis and draft manuscripts.

## Additional information

**Competing interests:** The authors declare no competing financial interests.

