## [Peer Review File · Nature Communications]

Reviewers' comments:

Reviewer #1 (Remarks to the Author):

It has long been a (reasonably held) assumption that a significant part of microbial community structure (in terms of nature and prevalence) is regulated by the exchange of extracellular metabolites (the 'exometabolome'). There has however been precious little general evidence for that. The attraction of this paper comes from the end statement of the abstract: "Our results demonstrate that metabolite profiling, sequencing and exometabolomics can be successfully integrated to functionally link metagenomes and microbial community structure with environmental chemistry." They also comment "To the best of our knowledge, this is the first study using isolate exometabolomics to link microbial community structure to soil chemistry."; it is also the first to MY knowledge too.

It also builds on the (to me anyway) possibly surprising discovery by this group to the effect that microbes really are rather more specialised (in terms of growth nutrients) than one might have expected, albeit this DOES perhaps more easily account for the enormous richness of the microbiomes. (They might care to make more of this.)

Overall, I liked the paper. It is a bit blow-by-blow, which is slightly inevitable given the voluminous data. However, figures such as Fig 2 can help summarise such data, and I'd encourage the authors to give thought to whether more data could be moved from the text to Figures.

Beyond this I do not have anything to say about the methods; they have been well established in this lab. I necessarily take the data for what they are. Metabolites were identified via authentic standards. Consequently, it might also be worth depositing them at Metabolights.

Some brief comments:

Table 1 and elsewhere. I know this is a bore, but linking the metabolite names either to name-independent representations (SMILES and/or InChI), or to persistent databases (PubChem, ChEBI) makes sure that folk know what they are, and also makes them much more computer-readable for folk who wish to do further analyses.

Line 179: Strikingly, of the 71 microbe-metabolite relationships evaluated (Supplementary Figure 6), 76% had the predicted directionality and would be very unlikely to occur by chance (two-tailed p value $< 1 \times 10^{-5}$; Supplementary Table 4). This seems rather pessimistic. If you had to predict 76% correctly of the elements of a binary string as being 1 or 0 the odds against this must surely be well less than that? May be worth rehearsing.

Line 259 "Here we find that unlike the boom-bust cycle of Firmicutes, the Actinobacteria phylum (such as *Blastococcus* sp.) may be more resistant to wetting." I do not think the subjunctive is necessary here. Sudden wetting creates a MASSIVE osmotic stress on organisms, and stretch-activated channels that release interior osmolytes are the best means to stop such bacteria from exploding! They underpin the many efflux systems for substances such as glutamate (important in commercial fermentations BY ACTIONBACTERIA) and accounts for the otherwise non-obvious property of an organism in excreting large amounts of biosynthesised nutrients! See e.g. Morbach S, Krämer R: Body shaping under water stress: osmosensing and osmoregulation of solute transport in bacteria. *ChemBioChem* 2002; 3:384-397; Kell DB, Swainston N, Pir P, Oliver SG: Membrane transporter engineering in industrial biotechnology and whole-cell biocatalysis. *Trends Biotechnol* 2015; 33:237-246.

I also agree with the comment: "One such advantage is that biocrust soil in this study is primarily quartz sand, facilitating metabolite analysis compared to many other soils which are typically rich in clays and other strongly-sorptive mineral surfaces."

Reviewer #2 (Remarks to the Author):

The paper by Swenson et al. assays exometabolite production and consumption in a soil crust community. The results confirm their previous study that suggested strong niche partitioning according to metabolic profiles among soil crust isolates and extends these findings to in situ

conditions by linking metabolite profiles, isolate characteristics, and metagenomes and - transcriptomes. Although the system chosen is relatively simple, the findings are impressive and important. The methodology (including statistical analysis) appear all sound and the conclusions are straightforward and do not overstate the case. I have only relatively minor comments.

Comments to consider:

My main concern is that the isolates used for comparison are not all that closely related to the populations occurring in situ. It is obvious that there is good agreement between the estimated dynamics of the microbial groups and the metabolites. As an explanation the authors cite the Martiny et al review claiming that "metabolic traits are largely conserved at the phylum level". That might be true for some traits but it is far from generally true and it may especially not be true for carbon utilization patterns. There can be great variation in these among very closely related populations. I suggest the authors explore the alternative explanation that the clades their isolates are part of have cohesive dynamics vs. the other clades considered. To me, this is an important point since much has been written recently about functional redundancy in microbial communities based largely on consideration of genes that can be easily annotated (often considering only the electron accepting half-reaction).

L.109: if they change why is it cycling?

L.129: Mention already here how distinct organisms are identified; what is the similarity cutoff used for taxonomic assignment?

Are organismal dynamics shown in Fig. 3 real data or conceptual?

Reviewer #3 (Remarks to the Author):

Review of "Linking soil biology and chemistry using bacterial isolate exometabolite profiles" by Swenson et al.

This manuscript is something of a proof-of-concept attempting to link soil microbial community composition with exometabolite dynamics. They examined the community composition (via metagenomics, using a ribosomal protein marker gene) and exometabolite composition from desert biological soil crusts at 4 different stages of development and 5 different time points after wetting. Based on cultures from the same crusts, they built a conceptual model of metabolite dynamics in the soil crusts by assuming that similar DNA sequences from the soil crusts are from organisms with essentially identical metabolisms and then attempted to provide support for this model from metatranscriptomic data.

While I'm enthusiastic about the concept of trying to link microbial community composition to exometabolites and thereby determine the linkage between the two—long a goal and a necessity in microbial ecology—I have significant concerns regarding the approach taken in this paper. My main concerns are that the manuscript overestimates the representativeness of their findings; they do not demonstrate a biological origin for their metabolites; and, perhaps most significantly, they do not convincingly demonstrate that their isolate data can be directly related to their environmental data. I explain these concerns in more detail below.

1. Do biological soil crusts represent all soils? The introduction to this manuscript seems to imply that soils are soils and that by studying dessicated and rehydrated soil crusts in a laboratory environment provides insights into between soil bacterial communities and biogeochemical transformations of soil organic carbon. I found the introductory material overly broad and difficult to justify as "linking SOM composition and microbial community structure".

2. The data are a bit hard to interpret due to a lack of proper controls. In a standard activity assay (which is essentially what the metabolomics characterization is), a no cell and killed cell control are

usually included; the first to demonstrate that the metabolites are not breaking down or forming spontaneously; the latter to examine the extent that leakage from dead cells or abiological transformations of metabolites by organic chemical reactions with dead biological material affects the outcomes. In this case, working with environmental samples, there can be no real “no cell” control, other than blanks made with the water utilized for rehydration (which they did). However, I’m quite concerned that there was no killed control. How do they know that the metabolites are biologically derived or being actively produced/consumed? This is particularly critical in this manuscript because they are attempting to associate metabolites with specific organisms. Furthermore, there appears to be no unwetted control to see if there’s an effect of light/dark cycles. Samples were taken from the field and stored in a dark dessication chamber, then rewetted and exposed to a light-dark cycle. Light can transform organic matter abiotically through photoreactions; such chemical and photo reactions remain unaccounted for in their system.

3. How long were samples “maintained in a dark dessication chamber”? Isn’t this likely to affect the metabolites, especially those formed soon after wetting? In particular, aren’t some components of the microbial system likely to be more sensitive to dessication than others, and more likely to lyse and release metabolites to the system? I realize that desert biological soil crusts are more likely to be inherently less sensitive to dessication than other soils; however, it’s unclear to me that dessication in a dessication chamber is equivalent to dessication in the field.

4. Using correlations from isolated microbes and then extrapolating them back to the environment is pretty iffy. I would assume that most of these metabolites, which I would not expect to be highly species specific due to their wide usage in general metabolism, are both produced and consumed by many members of the community. Furthermore, how an organism reacts in pure culture often is quite distinct from the way it reacts in a complex environment, both because the substrates available are quite different (growth media vs. soil organic matter) and because of positive and negative interactions within the microbial community. The author’s approach of assuming a positive correlation indicates a produced metabolite and a negative correlation indicates a consumed metabolite is oversimplified to the point of meaninglessness. This issue is further emphasized by the observation that these “relatively abundant” microbes represent a sum total of a maximum of ~35%, and a minimum of <5% of the total community (Figure S5). While these are relatively abundant, at least 2/3 of the community (and as much as 95% or greater of the community) is unaccounted for; to then build a model of metabolite transfer between these isolates and attempt to use this to explain metabolite patterns in the crusts seems a bridge too far to me.

5. How similar are the four isolates to the organisms that are the source of the sequences obtained from the crusts? Because they are focusing on exometabolites and the ability to utilize an exometabolite is likely to play important roles in fitness, it seems likely that genes and operons that play a role in producing/consuming these metabolites might well be transferred horizontally rather readily; therefore, marker genes may not be sufficient to relate the taxonomy of an organism to its physiology. Metagenomics could be used to help link the marker gene to functional genes; however, that was not done in this case despite them having the data set to do so. Furthermore, exometabolites are the product of physiological adaptations to changing environmental conditions. These physiological changes can, and do, happen at multiple levels—gene content/sequence, gene expression, protein expression, and post-translational modification, not to mention many other phenotypic features of cells such as lipid and carbohydrate content and composition, cell size, nucleic acid/other cellular component ratios, and many other features. Thus, even if two organisms are highly similar based on a marker gene sequence or even whole genome sequence, it is entirely possible that they would have entirely different metabolic and physiological responses to an environmental change, and that those different responses would wind up producing an entirely different exometabolite profile.

Although beyond the scope of this manuscript, some mechanism of distinguishing metabolic degradation and production pathways (other than genetic), such as isotopic labelling would have

made a clearer linkage.

Thus, while they have collected a fascinating data set that may have some implications for understanding the linkage between microbial community composition and soil carbon transformations, their model is not sufficiently supported by their data. The lack of complete controls and the jumps from laboratory-based to environmental data without supporting data to justify the links make it difficult to confidently support the proposed linkages.

Reviewer #4 (Remarks to the Author):

The authors of the manuscript "Linking soil biology and chemistry using bacterial isolate exometabolite profiles" propose an interesting approach where experimental data from soil metabolite profiling, metagenomics and transcriptomics were integrated to elucidate the population and environmental dynamics of biological crusts in soil gradient.

After reviewing their results, the findings are very interesting and novel, in such a way that can have a good impact in the scientific community. However, there is some extra analysis that could be performed and a few details that the authors should clarify before the manuscript could be published.

I list here my main concerns:

-The sequencing data for the metagenomics has not been made publicly available. As far as I remember, it should be available in a public repository such as the SRA (NCBI) or ENA (EBI) repositories. That would let other people reproduce the findings and support open access science.

-The taxonomic analysis was performed using the rplO genes which is a single copy gene marker. However, the authors complain about the low resolution of the marker. If they have whole metagenome shotgun information, it doesn't make sense to use just one marker. I suggest to use a program such as Kraken and repeat the analysis just to confirm that obtained from using rplO gene. Definitely, they will increase the resolution and reach species level instead of just family level as shown in Table 1. I understand that it was very convenient to use this marker since you can compare your results with previous ones but there should not be a problem to enrich and validate your current findings with a better analysis using the whole metagenome information to resolve at species level.

-The authors should make available all the in-house and customized scripts so people can reproduce their findings. About the metabolomics analysis, it is really a shame that the Metabolite Atlas database is not public. I'm not sure if they declare this limitation somewhere else but at least it is not written in the paper. Again, I'm not sure if this will break the openness policy of the journal.

-Please, include the dataset or results that were used in the "Transcriptomics support the link between soil microbes and metabolites" in Results section. It is very annoying to have to fetch another paper to check the current results. So, I would really appreciate if not only the results but the methods are included for this section in this paper.

Minor concerns:

-The sequencing was performed in the Illumina HiSeq 4000 platform. There is a duplication bias that has been reported here:

<https://sequencing.qcfail.com/articles/illumina-patterned-flow-cells-generate-duplicated-sequences/>

I think that this won't affect the analysis since the authors are not reporting the taxonomic abundance of all the community but just of certain organisms, but it would be good for them to know and consider it in case they want to perform other analysis.

-The Introduction section has no title "Introduction".

Reviewer #1 (Remarks to the Author):

We greatly appreciate your valuable comments and suggestions. To address these concerns, we have deposited data into the relevant online databases, clarified much of the text and added important information (e.g. metabolite representations in Supplementary Table 1A). We believe that these suggestions and related modifications have significantly improved the manuscript and further highlight the usefulness of our findings for the scientific community. Please see below for our point-by-point responses.

It has long been a (reasonably held) assumption that a significant part of microbial community structure (in terms of nature and prevalence) is regulated by the exchange of extracellular metabolites (the 'exometabolome'). There has however been precious little general evidence for that. The attraction of this paper comes from the end statement of the abstract: "Our results demonstrate that metabolite profiling, sequencing and exometabolomics can be successfully integrated to functionally link metagenomes and microbial community structure with environmental chemistry." They also comment "To the best of our knowledge, this is the first study using isolate exometabolomics to link microbial community structure to soil chemistry."; it is also the first to MY knowledge too.

It also builds on the (to me anyway) possibly surprising discovery by this group to the effect that microbes really are rather more specialised (in terms of growth nutrients) than one might have expected, albeit this DOES perhaps more easily account for the enormous richness of the microbiomes. (They might care to make more of this.)

Overall, I liked the paper. It is a bit blow-by-blow, which is slightly inevitable given the voluminous data. However, figures such as Fig 2 can help summarise such data, and I'd encourage the authors to give thought to whether more data could be moved from the text to Figures.

We appreciate the advice and based on this, realized that figure 3 was unclear. We now emphasize how this figure displays much of the data that is described in the text and we ensured that other described data are included in all figures and tables. For example, it is now emphasized that figure 3 represents real data (line 205):

"We next combined our correlation data with the isolate exometabolomics data in order to visualize a wetting-induced dynamic exometabolomic foodweb that may occur as a result of metabolite release by the primary producer (Microcoleus spp.) followed by heterotrophic consumption (in this specific case, the two heterotrophs that displayed sequential responses to wetting: Anoxybacillus sp. and Bacillus sp.) (Figure 3)."

Beyond this I do not have anything to say about the methods; they have been well established in this lab. I necessarily take the data for what they are. Metabolites were identified via authentic standards. Consequently, it might also be worth depositing them at Metabolights.

Metabolomics data have now been deposited at Metabolights with identifier information indicated in the "Data availability" section under Methods (line 539):

"Data availability

Sequencing data that support the findings of this study are available at the NCBI Sequence Read Archive (accession number PRJNA395099, <https://www.ncbi.nlm.nih.gov/bioproject/?term=PRJNA395099>) and metabolomics data have been deposited to the EMBL-EBI MetaboLights database (Haug et al, 2013) with the identifier MTBLS492 (<http://www.ebi.ac.uk/metabolights/MTBLS492>)."

Some brief comments:

Table 1 and elsewhere. I know this is a bore, but linking the metabolite names either to name-independent representations (SMILES and/or InChI), or to persistent databases (PubChem, ChEBI) makes sure that folk know what they are, and also makes them much more computer-readable for folk who wish to do further analyses.

This is a great point. Pubchem IDs and InChI keys have been added for each metabolite in Supplementary Table 1A. For example:

Supplementary Table 1. Metabolites detected in biocrust soil water.

Name	InChI Key	PubChem CID
urate	LEHOTFFKMJEONL-UHFF	1175
2'-deoxyguanosine	YKBGVTZYEHREMT-KVQE	187790
[2,3-dihydroxy-3-methylbutanoate]	JTEYKUFXXGOTEU-VKHM	440279
xanthine	LRFVYVWQMYALW-UHFF	1188
hypoxanthine	FDGQSTZIBFJUBT-UHFF	790
[lipote]	AGBQKNBQESQND-SSDC	6112
2'-deoxyuridine	MXHRCPNRJAMMIM-SHY	13712
urocanate	LOIYMIARKYCTBW-OWQ	736715
creatinine	DDRJAANPRJHGI-UHFF	588
stachyose	UQZYBXS HAGNOE-XNSR	439531
carnitine	PHIQXFUZVPPYI-ZCFIWI	10918
inosine	UGQMRVRMYASKQ-KQ	6021
nicotinate	PVNIIMVLHYAWGP-UHFF	938
4-hydroxy-proline	PMMYEEVYMWASQN-QV	440014
valine	KZSNWVFEVHDMF-BYP	6287
4-guanidinobutanoate	TUHVVEAJXIMEOSA-UHFF	500
uracil	ISAKRIDGNUMQOIC-UHFF	1174
guanine	UYTUPDQBNUYGX-UHF	764
leucine	ROHFNLRQFUQHCH-UHF	857
[alanyl-leucine]	RDIKFPRVJLJMER-BQBZG	96801
isoleucine	AGPKZVBTJNPAG-WHFB	6306
proline	ONIBWKKTOPOVIA-UHFF	614
thymine	RWQNRDOKXIBIV-UHFF	1135
[3-methyladenine]	FSASIHFSGAUM-UHFFA	1673
thymidine	IQFYKVMVGJFEH-XLPZG	5789
cytosine	OPTASPLRGRNAP-UHFF	597
disaccharide (maltose)	GUBGYTABKSRVRQ-PICCC	439186
phenylalanine	COLNVLDPVWVWLRQ-QMM	6140
betaine	KWIUHFFTVRNATP-UHFF	248
serine	MTCFGRXJLQNBG-REQ	5951
7-methyladenine	HCGYQLFMPXSDU-UHF	71593
threonine	AYFVYJQAPQTCCC-GBXIJ	6288
lysine	KDXKERNBIXSRK-YFKPB	5962
citrulline	RHGKLRLOHJIDR-BYPYZ	9750
adenine	GFFGJXBGBJISGV-UHFFA	190
salicylate	YGSDEFSMJLZEOE-UHFF	338
pyroglutamate	ODHCTXKNWHXJC-UHF	499
alanine	QNAYBMKLOCPYGI-REQ	5950

Line 179: Strikingly, of the 71 microbe-metabolite relationships evaluated (Supplementary Figure 6), 76% had the predicted directionality and would be very unlikely to occur by chance (two-tailed p value < 1 x 10⁻⁵; Supplementary Table 4). This seems rather pessimistic. If you had to predict 76% correctly of the elements of a binary string as being 1 or 0 the odds against this must sure be well less than that? May be worth rehearsing.

This sentence has been rephrased and clarified to read (line 200):

“Strikingly, of the 71 microbe-metabolite relationships evaluated (Supplementary Figure 7), 76% had the expected directionality that would be predicted based on isolate exometabolomics data. This overall observation of correct directionalities is significantly higher than what would be expected by chance.”

Line 259 “Here we find that unlike the boom-bust cycle of Firmicutes, the Actinobacteria phylum (such as *Blastococcus* sp.) may be more resistant to wetting.” I do not think the subjunctive is necessary here. Sudden wetting creates a MASSIVE osmotic stress on organisms, and stretch-activated channels that release interior osmolytes are the best means to stop such bacteria from exploding! They underpin the many efflux systems for substances such as glutamate (important in commercial fermentations BY ACTIONBACTERIA) and accounts for the otherwise non-obvious property of an organism in excreting large amounts of biosynthesised nutrients! See e.g. Morbach S, Krämer R: Body shaping under water stress: osmosensing and osmoregulation of solute transport in bacteria. *ChemBioChem* 2002; 3:384-397; Kell DB, Swainston N, Pir P, Oliver SG: Membrane

transporter engineering in industrial biotechnology and whole-cell biocatalysis. Trends Biotechnol 2015; 33:237-246.

We appreciate the insight and references. This section was initially unclear and was therefore rephrased to read (line 284):

“During biocrust wetting, we found that unlike the boom-bust cycle of Firmicutes, the Actinobacteria phylum (such as Blastococcus sp.) appeared more resistant to wetting duration.”

We also included the Morbach and Kell references in the discussion as suggested (line 297):

“Some likely biological mechanisms involved in the active metabolite “pulse” include osmotic stress induced release of solutes (e.g. proline, betaine and disaccharides) to maintain cell integrity^{41,42} as well as photosynthate release from the primary producer.”

I also agree with the comment: “One such advantage is that biocrust soil in this study is primarily quartz sand, facilitating metabolite analysis compared to many other soils which are typically rich in clays and other strongly-sorptive mineral surfaces.”

Reviewer #2 (Remarks to the Author):

We greatly appreciate your valuable comments and suggestions. To address these concerns, we have performed additional analyses to support our isolate-to-native microbial community comparisons including average nucleotide identity and average amino acid identity calculations. Furthermore, we have made several changes throughout the manuscript text emphasizing that these isolates may belong to clades with cohesive dynamics and clarifying our methods, results and discussion. We believe that these suggestions and related modifications have significantly improved the manuscript and further highlight the usefulness of our findings for the scientific community. Please see below for our point-by-point responses.

The paper by Swenson et al. assays exometabolite production and consumption in a soil crust community. The results confirm their previous study that suggested strong niche partitioning according to metabolic profiles among soil crust isolates and extends these findings to in situ conditions by linking metabolite profiles, isolate characteristics, and metagenomes and -transcriptomes. Although the system chosen is relatively simple, the findings are impressive and important. The methodology (including statistical analysis) appear all sound and the conclusions are straightforward and do not overstate the case. I have only relatively minor comments.

Comments to consider:

My main concern is that the isolates used for comparison are not all that closely related to the populations occurring in situ. It is obvious that there is good agreement between the estimated dynamics of the microbial groups and the metabolites. As an explanation the authors cite the Martiny et al review claiming that “metabolic traits are largely conserved at the phylum level”. That might be true for some traits but it is far from generally true and it may especially not be true for carbon utilization patterns. There can be great variation in these among very closely related populations. I suggest the authors explore the alternative explanation that the clades their isolates are part of have cohesive dynamics vs. the other clades considered. To me, this is an important point since much has been written recently about functional redundancy in microbial communities based largely on consideration of genes that can be easily annotated (often considering only the electron accepting half-reaction).

The reviewer brings up two excellent points, first, the relatedness between the isolates and the in situ populations and second, the alternative hypothesis that cohesiveness may underlie the observed dynamics.

To address the first point, we performed additional analysis to investigate the relatedness between our isolates and the in situ taxa using ANI and AAI calculations (Discussion, line 313):

“Next, we explored the connection between the observed microbe-metabolite relationships in biocrust and culture-based exometabolite profiles. Comparison of the two datasets was facilitated by the fact that the isolates were obtained from the same field site. Therefore, we used the phylogenetic marker, rplO (which had the most coverage in our dataset) to relate exometabolite-profiled isolates to native biocrust bacteria. Isolates and their closest matching environmental relatives ranged between 86-92% identical in their rplO sequence and 73-94% for ANI and 52-86% for AAI (Table 1). While there is a general lack of consensus of valid isolate-to-native population comparisons, our values indicate matching at approximately the genus or species level²⁸. Much of the difficulty in these comparisons is due to genomic heterogeneity within environmental samples⁴⁵. However, despite these complexities with determining exact phylogenetic distances, we observed functional similarity between the isolate and environmental exometabolomics datasets consistent with the view that many metabolic traits are conserved at the genus level⁴⁶.

The reviewer raises an interesting idea about the possible role of cohesiveness. We revised the manuscript to bring greater attention to these ideas and mention the possibility of cohesiveness in the discussion (line 328):

“An exciting alternative explanation for our observations is that the clades that these isolates are members of exhibit the same cohesive dynamics as the closest-matching bacteria in the biocrust⁴⁷.”

Ref 47: Herren, C.M. and McMahon, K.D. Cohesion: a method for quantifying the connectivity of microbial communities. The ISME Journal, 1-13, 2017.

L.109: if they change why is it cycling?

This is an excellent point. The term “cycling” has been removed and sentences rephrased appropriately (line 112):

“All 85 metabolites identified in the active biocrust samples changed at least two-fold (between minimum and maximum peak areas) across both wetting and successional stages (Figure 2).”

L.129: Mention already here how distinct organisms are identified; what is the similarity cutoff used for taxonomic assignment?

We now include a more thorough description of how biocrust community structure (and distinct organisms) were analyzed (line 131):

“Biocrust microbial community structure was inferred by shotgun metagenomics using a genome-centric pipeline²². In several recent reports²³⁻²⁶, ribosomal protein genes were used as phylogenetic markers from shotgun sequencing data (rather than the more classical 16S ribosomal RNA gene) because they exist as single copies in almost all genomes, assemble well from metagenome datasets (typically better than 16S), are well-conserved and have produced higher resolution phylogenetic trees²⁵. We identified a set of 17 previously-benchmarked single copy universal ribosomal protein genes²⁷ in our biocrust dataset and for community analysis, we selected rplO

(ribosomal protein L15) which had the most extensive community coverage based on the total number of assembled genes across our dataset (Supplementary Table 3).”

Are organismal dynamics shown in Fig. 3 real data or conceptual?

Thank you for pointing out that this is unclear. These dynamics are based on real data and is now indicated in the revised figure legend:

Figure 3. Simplified biocrust foodweb for three dominant biocrust bacteria based on combining isolate exometabolomics data with in situ microbe-metabolite relationships. This network displays the relationships between metabolites and three dominant bacteria as they increase and decrease in relative abundance across wetting and successional stages in biocrust. The lower line plot corresponds to real relative abundance measurements for the three bacteria in level C successional stage biocrust. As *Microcoleus* spp. increases in relative abundance immediately after wetting, many released metabolites (based on isolate exometabolomics, indicated by red arrows) are positively correlated with *Microcoleus* spp. in biocrust (solid red arrows) and as the two *Bacilli* increase in relative abundance (first *Anoxybacillus* sp. then *Bacillus* sp.), most consumed metabolites (indicated by blue arrows) decrease and are negatively correlated with these bacteria in biocrust (solid blue arrows) and most released metabolites are positively correlated (solid red arrows). Dotted arrows indicate metabolites that are released (red) or consumed (blue) that did not display the expected relationship with that microorganism in situ. The thickness of the line corresponds to the absolute value of the Spearman’s rho correlation coefficient.
* FDR < 0.05 for individual microbe-metabolite correlations. The expected directionality (solid lines versus dotted lines) was significant as determined by the exact binomial test ($p < 1 \times 10^{-5}$).

Reviewer #3 (Remarks to the Author):

We greatly appreciate your valuable comments and suggestions. To address these concerns, we have conducted additional supporting experiments showing that are our findings are biologically-driven (rather than due to abiotic processes). We also performed additional analyses to support our isolate-to-native microbial community comparisons including average nucleotide identity and average amino acid identity calculations. In response to your comments we have made several changes throughout the manuscript text highlighting the specificity of these findings to the biocrust ecosystem and clarifying our methods, results and discussion. We believe that together these suggestions and related modifications have significantly improved the manuscript and further highlight the usefulness of our findings for the scientific community. Please see below for our point-by-point responses.

Review of “Linking soil biology and chemistry using bacterial isolate exometabolite profiles” by Swenson et al.

This manuscript is something of a proof-of-concept attempting to link soil microbial community composition with exometabolite dynamics. They examined the community composition (via metagenomics, using a ribosomal protein marker gene) and exometabolite composition from desert biological soil crusts at 4 different stages of development and 5 different time points after wetting. Based on cultures from the same crusts, they built a conceptual model of metabolite dynamics in the soil crusts by assuming that similar DNA sequences from the soil crusts are from organisms with essentially identical metabolisms and then attempted to provide support for this model from metatranscriptomic data.

While I’m enthusiastic about the concept of trying to link microbial community composition to

exometabolites and thereby determine the linkage between the two—long a goal and a necessity in microbial ecology—I have significant concerns regarding the approach taken in this paper. My main concerns are that the manuscript overestimates the representativeness of their findings; they do not demonstrate a biological origin for their metabolites; and, perhaps most significantly, they do not convincingly demonstrate that their isolate data can be directly related to their environmental data. I explain these concerns in more detail below.

Thank you for these very helpful comments. These concerns have been addressed to the best of our ability by conducting further experiments and analyses as indicated below.

1. Do biological soil crusts represent all soils? The introduction to this manuscript seems to imply that soils are soils and that by studying dessicated and rehydrated soil crusts in a laboratory environment provides insights into between soil bacterial communities and biogeochemical transformations of soil organic carbon. I found the introductory material overly broad and difficult to justify as “linking SOM composition and microbial community structure”.

We have revised the text to clarify that these are not representative of all soils. We have added “biological soil crust” to emphasize this point and the title now reads:

*“Linking soil biology and chemistry in **biological soil crust** using isolate exometabolomics”.*

The introduction has also been modified to emphasize that this is specific to biocrust and that this system was selected because it is a relatively ‘simple’ soil system and for its tractability and unique ability to undergo lab-based microbial and metabolic dynamics. For example (line 83):

“Here we exploited the dynamic and tractable characteristics of biocrust to explore the relationships between soil microbes and metabolites (for this particular ecosystem) and then determined the extent to which isolate exometabolite patterns are conserved in situ (within the intact biocrust soil community).”

2. The data are a bit hard to interpret due to a lack of proper controls. In a standard activity assay (which is essentially what the metabolomics characterization is), a no cell and killed cell control are usually included; the first to demonstrate that the metabolites are not breaking down or forming spontaneously; the latter to examine the extent that leakage from dead cells or abiological transformations of metabolites by organic chemical reactions with dead biological material affects the outcomes. In this case, working with environmental samples, there can be no real “no cell” control, other than blanks made with the water utilized for rehydration (which they did). However, I’m quite concerned that there was no killed control. How do they know that the metabolites are biologically derived or being actively produced/consumed? This is particularly critical in this manuscript because they are attempting to associate metabolites with specific organisms. Furthermore, there appears to be no unwetted control to see if there’s an effect of light/dark cycles. Samples were taken from the field and stored in a dark dessication chamber, then rewetted and exposed to a light-dark cycle. Light can transform organic matter abiotically through photoreactions; such chemical and photo reactions remain unaccounted for in their system.

To address this important point, we performed additional experimentation including both no-cell and killed controls. The no-cell control (water, no soil) was included which shows that the metabolites analyzed were not produced by abiotic mechanisms (such as leaching from containers). The killed control is described in the methods, results, Supplementary Table 1B, Supplementary Table 2 and Supplementary Figure 3:

Methods (line 440):

“An extraction control (three replicates of water, no soil) was included and sampled at 49.5 h to evaluate background contamination. To control for abiotic metabolite processes (such as sorption, photodegradation, etc), a separate experiment was performed where soil water was sampled along the same timepoints (5 sampling timepoints in triplicate) from killed (autoclaved four times) biocrust from a late successional stage.”

Methods (line 485):

“Significance between temporal patterns from active biocrust and killed control samples was analyzed using the anovan and multcompare functions in Matlab R2016A with an alpha of 0.05 corresponding to 95% confidence level and Tukey's honestly significant difference test where time is considered a continuous variable and succession is considered a categorical variable (Supplementary Table 2).”

Results (line 126):

“These dynamics, especially metabolite consumption, were largely of biological origin as demonstrated by the killed controls where the temporal patterns for most (53 out of 72) metabolites were significantly different ($p < 0.05$) than the active biocrust samples (Supplementary Tables 1B and 2; Supplementary Figure 3).”

Supplementary Figure 3:

colorkeys

heatmap

row groups

Supplementary Figure 3. Metabolite dynamics observed in biocrust soil water from active samples (early to late successional stages) compared to killed control samples. The heatmap displays changes in metabolite peak area (log₂ fold change) relative to the first sampling timepoint (3 min) within each successional stage and within the killed control dataset for the 72 metabolites detected across all five datasets. Putative metabolites are indicated by parentheses. $n = 2-5$ for each treatment. * $p < 0.05$ for metabolites that are significantly different in temporal patterns between the killed control and at least one successional stage of active biocrust (Supplementary Table 2).

3. How long were samples “maintained in a dark dessication chamber”? Isn’t this likely to affect the metabolites, especially those formed soon after wetting? In particular, aren’t some components of the microbial system likely to be more sensitive to dessication than others, and more likely to lyse and release metabolites to the system? I realize that desert biological soil crusts are more likely to be inherently less sensitive to dessication than other soils; however, it’s unclear to me that dessication in a dessication chamber is equivalent to dessication in the field.

Thank you for pointing out that this was unclear. Dry samples were collected in the field and maintained in this desiccated state for 11 months prior to this study—an approach that we and other labs have used in many other studies. Importantly, we do not dry these samples in the lab for the reason the reviewer states. This is now described more clearly in the methods (line 423):

“Dry samples collected in the field were maintained in a dark desiccation chamber in the laboratory for approximately 11 months until lab-based experiments were performed consistent with previous reports [Strauss et al, 2011]. Biocrusts are adapted to prolonged desiccation though it is important to note that the duration of desiccation likely affects metabolites composition and biological responses.”

4. Using correlations from isolated microbes and then extrapolating them back to the environment is pretty iffy. I would assume that most of these metabolites, which I would not expect to be highly species specific due to their wide usage in general metabolism, are both produced and consumed by many members of the community. Furthermore, how an organism reacts in pure culture often is quite distinct from the way it reacts in a complex environment, both because the substrates available are quite different (growth media vs. soil organic matter) and because of positive and negative interactions within the microbial community. The author’s approach of assuming a positive correlation indicates a produced metabolite and a negative correlation indicates a consumed metabolite is oversimplified to the point of meaninglessness.

This issue is further emphasized by the observation that these “relatively abundant” microbes represent a sum total of a maximum of ~35%, and a minimum of <5% of the total community (Figure S5). While these are relatively abundant, at least 2/3 of the community (and as much as 95% or greater of the community) is unaccounted for; to then build a model of metabolite transfer between these isolates and attempt to use this to explain metabolite patterns in the crusts seems a bridge too far to me.

The reviewer brings up three excellent points regarding 1) metabolite specificity to an organism; 2) bacterial functions in pure culture versus the environment and 3) modeling based on 4 bacteria.

1) Metabolite specificity to an organism: We agree that these metabolites are likely to be consumed by many organisms, though we expect the most abundant organisms will dominate metabolic dynamics and have included additional discussion accordingly. We now include this (line 346):

“We next used the biocrust microbe-metabolite relationships to display a simplified dynamic exometabolomics web describing the wetting response of three dominant bacteria in biocrust (Figure 3). While this network portrays the dynamic substrate preferences of three specific native biocrust bacteria, one can imagine that there are many functionally-similar microorganisms that could fall into the categories of ‘early responders’, ‘mid-responders’ and ‘late-responders’ for a wetting event.”

2) Bacterial functions in pure culture vs. the environment: Regarding the reviewer’s second point, organisms in culture may indeed behave differently than in the environment. However, to recapitulate environmental dynamics as much as possible, our isolate-based study used a biocrust-relevant media to examine exometabolite profiles of biocrust isolates. One such medium contained *M. vaginatus* extracts based on the assumption that this organism is the primary producer in biocrust. Despite these inherent drawbacks in comparing the two environments (lab vs. in situ), we were excited to discover that there was some agreement between the datasets. The relatedness of the cultures to environment is now mentioned in the introduction (line 92):

“While the comparison of a microbe in isolation and in an environmental system is challenging, isolate exometabolomics were performed with media supplemented with lysed cell metabolite extracts to simulate the biocrust environment²⁰”.

3) Modeling based on 4 bacteria: We agree that modeling based on a few microbes would be problematic and apologize that the manuscript was not clear on this point. Figure 3 presents correlations (data) between microbes and metabolites and is not a model. We agree that for some samples, these 4 bacteria don't make up a large fraction of the community, and now address the possibility that these dynamics may reflect the collective behavior of functionally-similar organisms (as shown in the first point above). Figure 3 is now clarified in the manuscript (line 205):

“We next combined our correlation data with the isolate exometabolomics data in order to visualize a wetting-induced dynamic exometabolomic foodweb that may occur as a result of metabolite release by the primary producer (Microcoleus spp.) followed by heterotrophic consumption (in this specific case, the two heterotrophs that displayed sequential responses to wetting: Anoxybacillus sp. and Bacillus sp.) (Figure 3).”

5. How similar are the four isolates to the organisms that are the source of the sequences obtained from the crusts? Because they are focusing on exometabolites and the ability to utilize an exometabolite is likely to play important roles in fitness, it seems likely that genes and operons that play a role in producing/consuming these metabolites might well be transferred horizontally rather readily; therefore, marker genes may not be sufficient to relate the taxonomy of an organism to its physiology. Metagenomics could be used to help link the marker gene to functional genes; however, that was not done in this case despite them having the data set to do so. Furthermore, exometabolites are the product of physiological adaptations to changing environmental conditions. These physiological changes can, and do, happen at multiple levels—gene content/sequence, gene expression, protein expression, and post-translational modification, not to mention many other phenotypic features of cells such as lipid and carbohydrate content and composition, cell size, nucleic acid/other cellular component ratios, and many other features. Thus, even if two organisms are highly similar based on a marker gene sequence or even whole genome sequence, it is entirely possible that they would have entirely different metabolic and physiological responses to an environmental change, and that those different responses would wind up producing an entirely different exometabolite profile.

Since the metabolites analyzed here are so general, functional genes unfortunately did not provide much information in this case. Therefore, we calculated the average nucleotide identity and average amino acid identity between isolates and their related environmental organisms to show that these are the closest environmental relatives. These explanations in the discussion can be found in the paragraph starting at line 313:

“Next, we explored the connection between the observed microbe-metabolite relationships in biocrust and culture-based exometabolite profiles. Comparison of the two datasets was facilitated by the fact that the isolates were obtained from the same field site. Therefore, we used the phylogenetic marker, rplO (which had the most coverage in our dataset) to relate exometabolite-profiled isolates to native biocrust bacteria. Isolates and their closest matching environmental relatives ranged between 86-92% identical in their rplO sequence and 73-94% for ANI and 52-86% for AAI (Table 1). While there is a general lack of consensus of valid isolate-to-native population comparisons, our values indicate matching at approximately the genus or species level²⁸. Much of the difficulty in these comparisons is due to genomic heterogeneity within environmental samples⁴⁵. However, despite these complexities with determining exact phylogenetic distances, we observed functional similarity between the isolate and environmental exometabolomics datasets consistent with the view that many metabolic traits are conserved at the genus level⁴⁶. An exciting alternative explanation for our observations is that the clades that these isolates are members of exhibit the same cohesive dynamics as the closest-matching bacteria in the biocrust⁴⁷.”

Although beyond the scope of this manuscript, some mechanism of distinguishing metabolic degradation and production pathways (other than genetic), such as isotopic labelling would have

made a clearer linkage.

We agree- that is a great future experimental plan!

Thus, while they have collected a fascinating data set that may have some implications for understanding the linkage between microbial community composition and soil carbon transformations, their model is not sufficiently supported by their data. The lack of complete controls and the jumps from laboratory-based to environmental data without supporting data to justify the links make it difficult to confidently support the proposed linkages.

We appreciate all of the reviewer's valuable comments and suggestions and based on these, we believe that we have now provided adequate controls and isolate-to-environmental genome comparative analyses (and discussion points) to provide some much-needed links between phylogeny and function. We also hope data and approaches such as ours can be used as constraints in future carbon cycling modeling.

Reviewer #4 (Remarks to the Author):

We greatly appreciate your valuable comments and suggestions. To address these concerns, we have deposited data and code into the relevant online databases, more clearly explained our selection of *rplo* as a phylogenetic marker, performed additional analyses to support our isolate-to-native microbial community comparisons including average nucleotide identity and average amino acid identity calculations and clarified much of the text and added important information (e.g. relevant *Microcoleus* gene expression data). We believe that these suggestions and related modifications have significantly improved the manuscript and further highlight the usefulness of our findings for the scientific community. Please see below for our point-by-point responses.

The authors of the manuscript "Linking soil biology and chemistry using bacterial isolate exometabolite profiles" propose an interesting approach where experimental data from soil metabolite profiling, metagenomics and transcriptomics were integrated to elucidate the population and environmental dynamics of biological crusts in soil gradient.

After reviewing their results, the findings are very interesting and novel, in such a way that can have a good impact in the scientific community. However, some extra analysis could be performed and a few details that the authors should clarify before the manuscript could be published.

We thank the referee for their effort in reviewing this and their enthusiasm for this study.

I list here my main concerns:

-The sequencing data for the metagenomics has not been made publicly available. As far as I remember, it should be available in a public repository such as the SRA (NCBI) or ENA (EBI) repositories. That would let other people reproduce the findings and support open access science.

These data are now publicly available in NCBI SRA as indicated in the "data availability" section of the manuscript (line 539):

"Sequencing data that support the findings of this study are available at the NCBI Sequence Read Archive (accession number PRJNA395099, <https://www.ncbi.nlm.nih.gov/bioproject/?term=PRJNA395099>) and metabolomics data have been deposited to the EMBL-EBI MetaboLights database⁷¹ with the identifier MTBLS492

(<http://www.ebi.ac.uk/metabolights/MTBLS492>).”

-The taxonomic analysis was performed using the *rpI*O genes which is a single copy gene marker. However, the authors complain about the low resolution of the marker. If they have whole metagenome shotgun information, it doesn't make sense to use just one marker. I suggest to use a program such as Kraken and repeat the analysis just to confirm that obtained from using *rpI*O gene. Definitely, they will increase the resolution and reach species level instead of just family level as shown in Table 1. I understand that it was very convenient to use this marker since you can compare your results with previous one but there should not be a problem to enrich and validate your current finding with a better analysis using the whole metagenome information to resolve at species level.

Thank you for the advice. After further analysis of other phylogenetic markers, we found that they did not have as adequate coverage as the *rpI*O gene (Supplementary Table 3). This is now described in the results (line 132):

*“In several recent reports²³⁻²⁶, ribosomal protein genes were used as phylogenetic markers from shotgun sequencing data (rather than the more classical 16S ribosomal RNA gene) because they exist as single copies in almost all genomes, assemble well from metagenome datasets (typically better than 16S), are well-conserved and have produced higher resolution phylogenetic trees²⁵. We identified a set of 17 previously-benchmarked single copy universal ribosomal protein genes²⁷ in our biocrust dataset and for community analysis, we selected *rpI*O (ribosomal protein L15) which had the most extensive community coverage based on the total number of assembled genes across our dataset (Supplementary Table 3).”*

Supplementary Table 3:

gene	alt. name	gene counts
L2	rpIB	434
L3	rpIC	402
L4	rpID	436
L5	rpIE	450
L6	rpIF	447
L10	rpIJ	430
L14	rpIN	401
L15	rpIO	466
L16	rpIP	407
L18	rpIR	401
L22	rpIV	348
L24	rpIX	425
S3	rpsC	375
S8	rpsH	461
S10	rpsJ	350
S17	rpsQ	412
S19	rpsS	355

As additional support for our *rpI*O-based results, we performed additional analysis to investigate the relatedness between our isolates and the in situ taxa using ANI and AAI calculations (Line 313):

*“Next, we explored the connection between the observed microbe-metabolite relationships in biocrust and culture-based exometabolite profiles. Comparison of the two datasets was facilitated by the fact that the isolates were obtained from the same field site. Therefore, we used the phylogenetic marker, *rpI*O (which had the most coverage in our dataset) to relate exometabolite-profiled isolates*

to native biocrust bacteria. Isolates and their closest matching environmental relatives ranged between 86-92% identical in their rplO sequence and 73-94% for ANI and 52-86% for AAI (Table 1). While there is a general lack of consensus of valid isolate-to-native population comparisons, our values indicate matching at approximately the genus or species level²⁸. Much of the difficulty in these comparisons is due to genomic heterogeneity within environmental samples⁴⁵. However, despite these complexities with determining exact phylogenetic distances, we observed functional similarity between the isolate and environmental exometabolomics datasets consistent with the view that many metabolic traits are conserved at the genus level⁴⁶. An exciting alternative explanation for our observations is that the clades that these isolates are members of exhibit the same cohesive dynamics as the closest-matching bacteria in the biocrust⁴⁷.”

-The authors should make available all the in-house and customized scripts so people can reproduce their findings. About the metabolomics analysis, it is really a shame that the Metabolite Atlas database is not public. I'm not sure if they declare this limitation somewhere else but at least it is not written in the paper. Again, I'm not sure if this will break the openness policy of the journal.

Scripts used to analyze metabolomics data (Metabolite Atlas) are public and the link is now provided under “Code Availability” in the methods section (line 547):

“Metabolite atlas can be accessed at <https://github.com/biorack/metatlas>.”

-Please, include the dataset or results that were used in the "Transcriptomics support the link between soil microbes and metabolites" in Results section. It is very annoying to have to fetch another paper to check the current results. So, I would really appreciate if not only the results but the methods are included for this section in this paper.

We apologize for the inconvenience. All relevant information is included in the methods and Supplementary Table 7.

Methods, Line 531:

Microcoleus gene expression analysis

Microcoleus genes from Supplementary Table S6 from Rajeev et al (2013) were categorized into KEGG pathways (for genes containing KEGG ID numbers), which are summarized in Supplementary Table 6. Analyses focused on pathways that are primarily anabolic or catabolic for metabolites that were released by Microcoleus PCC 9802. The average fold-change (relative to dry biocrusts) and standard errors were calculated for all genes belonging to pathways of interest.

Example of Supp. Table 7:

Supplementary Table 7. Data obtained from Rajeev et al (2013) and categorized into relevant KEGG pathways. Values indicate fold change relative to dry biocrust (time 0).

BIOSYNTHESIS

Biosynthesis of amino acids

SEQ ID	KEGG ID	GENE_INF	Time 0	3 min	15 min	1 hr	9 hr	11.5 hr	18 hr	22 hr	25.5 hr	31.5 hr	35.5 hr	42 hr	4
2505166495	K01703	2505166495	1.000	1.389	0.876	1.404	1.298	1.745	1.618	1.559	1.726	1.782	1.719	1.635	
2505165971	K15634	2505165971	1.000	1.359	1.503	1.633	2.940	3.842	2.966	2.899	3.955	4.491	3.680	4.097	
2505168303	K01940	2505168303	1.000	1.259	1.086	1.807	3.323	4.019	4.995	4.671	6.661	6.705	4.686	7.226	
2505169860	K00053	2505169860	1.000	1.245	2.262	2.790	7.698	9.630	7.515	6.601	8.782	9.695	6.282	8.315	
2505168983	K00768	2505168983	1.000	1.186	0.959	1.062	1.339	1.349	1.422	1.404	1.880	1.923	1.366	1.522	
2505167806	K00286	2505167806	1.000	1.158	1.351	1.103	1.518	1.381	1.123	1.208	1.285	1.098	1.131	1.227	
2505169947	K10206	2505169947	1.000	1.157	1.543	1.196	2.002	1.784	1.487	1.578	1.749	1.718	1.331	1.465	
2505166565	K10206	2505166565	1.000	1.156	0.910	0.953	1.882	1.696	1.483	1.207	1.748	1.700	1.492	1.508	
2505168309	K01647	2505168309	1.000	1.127	1.095	1.285	1.489	1.537	1.318	1.475	1.586	1.876	1.425	1.559	
2505168407	K00600	2505168407	1.000	1.094	0.505	0.870	2.042	2.481	1.760	1.402	2.588	2.623	1.874	1.807	
2505166246	K00826	2505166246	1.000	1.087	1.053	0.976	1.980	1.871	2.132	1.815	2.462	2.677	1.373	2.237	
2505171087	K01658	2505171087	1.000	1.086	1.229	1.062	1.401	1.446	1.432	1.897	1.656	1.616	1.359	1.735	
2505168845	K00266	2505168845	1.000	1.069	0.470	0.867	1.454	1.586	1.206	1.016	1.566	1.975	1.454	1.117	
2505170664	K00615	2505170664	1.000	1.059	0.980	1.820	1.773	1.999	2.686	2.473	2.815	2.730	1.856	2.792	
2505169287	K01915	2505169287	1.000	1.023	0.912	2.605	2.820	3.965	4.085	4.867	6.813	5.070	5.185	5.678	
2505168569	K00031	2505168569	1.000	1.016	1.006	1.316	2.402	3.991	3.436	3.245	3.111	3.705	2.683	3.016	
2505167522	K01649	2505167522	1.000	0.995	1.070	1.517	1.636	2.034	2.097	1.718	1.868	2.023	2.070	2.153	
2505170763	K00818	2505170763	1.000	0.980	0.834	1.724	2.294	2.218	2.209	2.042	3.234	4.301	3.048	3.417	
2505169225	K01834	2505169225	1.000	0.965	1.365	2.245	6.256	7.197	9.998	5.715	5.489	10.078	5.817	6.267	
2505170019	K15634	2505170019	1.000	0.945	0.879	1.380	1.591	2.327	2.083	1.975	1.748	1.742	1.413	1.863	
2505166395	K01889	2505166395	1.000	0.944	0.597	0.663	1.118	1.826	1.112	0.973	1.373	2.282	1.632	1.575	
2505166396	K00145	2505166396	1.000	0.938	0.857	1.596	1.515	2.129	2.238	2.093	2.858	2.586	2.245	2.711	
2505167133	K01915	2505167133	1.000	0.931	0.588	3.120	6.176	6.326	5.387	4.574	9.090	10.982	6.445	5.806	
2505170540	K01652	2505170540	1.000	0.930	0.750	1.514	2.510	2.919	2.489	2.543	1.803	2.389	2.123	3.076	
2505165612	K01649	2505165612	1.000	0.886	0.822	1.713	1.786	2.664	2.180	2.140	2.324	2.222	3.108	2.182	
2505170503	K00930	2505170503	1.000	0.878	0.930	1.418	2.185	2.335	2.903	2.969	3.325	3.437	2.273	3.529	
2505171067	K01657	2505171067	1.000	0.871	0.976	1.420	1.646	2.539	3.435	2.625	2.916	1.895	1.943	3.213	
2505171024	K01687	2505171024	1.000	0.861	0.854	1.208	1.498	1.453	1.073	1.137	1.053	1.318	1.141	1.132	
2505167835	K00058	2505167835	1.000	0.860	0.880	1.569	2.252	3.140	2.411	2.266	3.290	3.560	3.378	3.098	
2505166234	K00927	2505166234	1.000	0.849	0.726	0.900	1.178	1.401	1.634	1.561	1.584	1.287	0.856	1.744	
2505170982	K00616	2505170982	1.000	0.835	1.046	0.952	1.307	2.022	1.729	1.458	1.255	1.275	1.733	1.400	
2505170636	K00134	2505170636	1.000	0.823	1.094	1.588	2.734	2.707	1.302	1.193	2.006	4.255	3.099	1.792	

Minor concerns:

-The sequencing was performed in the illumina HiSeq 4000 platform. There is a duplication bias that has been reported here:

<https://sequencing.qcfail.com/articles/illumina-patterned-flow-cells-generate-duplicated-sequences/>

I think that this won't affect the analysis since the authors are not reporting the taxonomic abundance of all the community but just of certain organisms, but it would be good for them to know and consider it in case they want to perform other analysis.

We appreciate the information and agree that this shouldn't affect the analysis. Thanks!

-The Introduction section has no title "Introduction".

This has been added.

Reviewer #2 (Remarks to the Author):

I thank the reviewers for addressing my comments. I fully realize how hard linking the activity with identity in experimental environmental work and I think the authors have done a satisfactory job.

One remaining comment is that it would be very odd if the AAI values were consistently lower than the ANI values. Please redo your calculations.

Reviewer #3 (Remarks to the Author):

In general, the authors have done a commendable job addressing some rather complex critiques I and the other reviewers provided on the original manuscript. Most of my concerns have been addressed in this revised version, and I am particularly impressed with the inclusion of the killed controls and the acknowledgement of that important component of the experiment in the text.

However, I have several concerns remaining:

1. While they do include information on the killed control, they note that for 53 of 72 metabolites, they differ significantly (with a $p < 0.05$) from the live samples. It is not clear to me what the situation is with the remaining 19 metabolites--were these then excluded from subsequent analysis? If not, how do they know whether the changes observed are from biological or abiological activity?

Also, in line 112 they state that there are 85 identified metabolites; however, in line 128 they discuss "53 out of 72" metabolites--I'm not clear about this. Were there fewer metabolites in the killed controls? Of the metabolites in the killed controls, did they overlap 100% with those from the live samples? This information is available in the supplementary information, but should be included in the text.

2. It's still not clear to me why they use a single gene only for their phylogenetic analysis. They have binned genome sequences; they should be able to utilize multiple genes and see how congruent the phylogenies are. Using a single gene to show the level of similarity between their isolates and the sequences from the metagenome seems problematic to me--even if ANI data are included.

3. While they have done a good job in the discussion placing caveats around their interpretation of the exometabolites coming from or being consumed by their specific isolates, their results and Figure 3 are not nearly so careful in terms of language or clarification regarding their confidence that these isolates are truly representative of their nearest neighbours in the crusts. I still think that they need to tone down the language a bit, and to clarify that figure 3 is a conceptual model and that those organisms labeled on the figure may be representative of metabolic groups.

Overall, my impression is that the manuscript is greatly improved but that it still suffers from a bit of overinterpretation of some data and incomplete analysis of other data.

Reviewer #4 (Remarks to the Author):

Here are my comments to the reply of the authors:

'These data are now publicly available in NCBI SRA as indicated in the "data availability" section of

the manuscript (line 539):

"Sequencing data that support the findings of this study are available at the NCBI Sequence Read Archive (accession number PRJNA395099, <https://www.ncbi.nlm.nih.gov/bioproject/?term=PRJNA395099>) and metabolomics data have been deposited to the EMBL-EBI MetaboLights database⁷¹ with the identifier MTBLS492 (<http://www.ebi.ac.uk/metabolights/MTBLS492>)."

The data is now at the NCBI. Just check one of the biosamples which seems to be repeated and has no SRA data linked to it.

<https://www.ncbi.nlm.nih.gov/biosample/7370990>

'As additional support for our rplO-based results, we performed additional analysis to investigate the relatedness between our isolates and the in situ taxa using ANI and AAI calculations (Line 313):'

I do not agree with this analysis. The authors used the ANI (average nucleotide identity) and AAI (average aminoacid identity) but their results don't make sense:

'Isolates and their closest matching environmental relatives ranged between 86-92% identical in their rplO sequence and 73-94% for ANI and 52-86% for AAI (Table 1). While there is a general lack of consensus of valid isolate-to-native population comparisons, our values indicate matching at approximately the genus or species level'

In the case of ANI, it make more sense to use it as a genus or species identification parameter only when you have long DNA fragments like contigs. I don't understand how you get a higher ANI value than the AAI, it should be the other way around.

Also, the authors are missing the point. If they really want to obtain more information about the taxonomy, they should use all the information contained in the shotgun reads. As I suggested, they can use a k-mer spectra based analysis such as CLARK or Kraken that are really easy to use and can be run in less than an hour. Alternatively, if they want to use single copy markers, they can use ANPHORA2 or MOCAT or MetaPhlAn. These programs use several single copy gene markers.

I don't see the point on doing an analysis with just one marker when you have much more information...

'Scripts used to analyze metabolomics data (Metabolite Atlas) are public and the link is now provided under "Code Availability" in the methods section (line 547): "Metabolite atlas can be accessed at <https://github.com/biorack/metatlas>."

Several of the links (Installation, Examples and Information) inside the project web page are broken. Also, they completely neglected my comment about the restricted access to Metabolite Atlas. Even if now the software to analyze the LCMS data, there is no access to the database.

My impression is that this is a software that will work at the National Energy Research Scientific Computing Center (NERSC) but not somewhere else...

Reviewers' comments:

Reviewer #2 (Remarks to the Author):

We appreciate the reviewer's feedback on our manuscript revisions. In response to their concern, we updated the text to focus on ANI and clarify that this value is based on whole genome comparisons while AAI is based on gene comparison (which is supported by an included reference). Please see below for more details.

I thank the reviewers for addressing my comments. I fully realize how hard linking the activity with identity in experimental environmental work and I think the authors have done a satisfactory job.

One remaining comment is that it would be very odd if the AAI values were consistently lower than the ANI values. Please redo your calculations.

We thank the reviewer for pointing this out, which is something we needed to clarify in the manuscript. Here, ANI was calculated based as "whole genome" average nucleotide identity and AAI as "gene" average amino acid identity (for coding loci). In a reference provided within the text (Rodriguez-R and Konstantinidis, 2014) they show that within the context of cultivation-free identification of bacterial species (within/between species/genus) AAI is in most cases is smaller than ANI (refer to Figure 3 within the reference which shows the comparison of AAI and ANI for complete bacterial genomes stratified by taxonomic names (species, genus, and division)). Since ANI is more widely used and to avoid confusion, these values were kept in the main text (Table 1) and AAI was moved to a supplementary table (5).

These few additional details are now included in the text (line 168):

*“Further comparative analyses were conducted to calculate genome average nucleotide identity (ANI) between isolate genomes and their related metagenome-assembled genomes (MAGs) to which the *rpIO* gene was co-binned, which validated that each isolate and their closest matching environmental relative were of the same genus or species following the conventions described in a previous report²⁸ (Table 1). Additional validation was obtained by comparing gene average amino acid identity (AAI) (Supplementary Table 5) and other ribosomal protein genes from within each MAG to the corresponding isolate (Supplementary Table 6).”*

Reviewer #3 (Remarks to the Author):

We appreciate the valuable feedback and suggestions from the reviewer. To address these concerns, we removed ambiguity regarding biotic/ abiotic controls on metabolite dynamics for the isolate exometabolomics comparison by excluding metabolites that were not significantly different between the two datasets, did a more thorough gene-gene comparison between isolate genomes and environmental metagenomes using single-copy phylogenetic markers and clarified sections of the text to reduce over-interpretation of data. Please see below for details on these points.

In general, the authors have done a commendable job addressing some rather complex critiques I and the other reviewers provided on the original manuscript. Most of my concerns

have been addressed in this revised version, and I am particularly impressed with the inclusion of the killed controls and the acknowledgement of that important component of the experiment in the text.

However, I have several concerns remaining:

1. While they do include information on the killed control, they note that for 53 of 72 metabolites, they differ significantly (with a $p < 0.05$) from the live samples. It is not clear to me what the situation is with the remaining 19 metabolites--were these then excluded from subsequent analysis? If not, how do they know whether the changes observed are from biological or abiological activity?

Also, in line 112 they state that there are 85 identified metabolites; however, in line 128 they discuss "53 out of 72" metabolites--I'm not clear about this. Were there fewer metabolites in the killed controls? Of the metabolites in the killed controls, did they overlap 100% with those from the live samples? This information is available in the supplementary information, but should be included in the text.

We thank the reviewer for pointing this out. As suggested, for isolate exometabolomics comparisons, we removed non-significant metabolites resulting in 69% of the microbe-metabolite relationships being consistent with our isolate exometabolomics data and revised the results and figure 3 accordingly. Results (line 229):

"Of the 85 metabolites identified in the biocrust soil water, 32 matched the isolate exometabolome dataset. Nine of these displayed temporal patterns that were not significantly different than the killed control and were excluded from this analysis since abiotic controls on these could not be ruled out (Supplementary Table 7)."

This new analysis led to a completely re-done (and simplified) Figure 3. The new results appear to have a negligible impact on our overall conclusions:

Regarding the second comment, the reviewer is correct: Fewer metabolites were detected in the killed controls, likely because those are produced by biological mechanisms which is now more explicitly stated in the results (line 129):

“These dynamics, especially metabolite consumption, were largely of biological origin as demonstrated by comparison with the killed controls. Thirteen metabolites that were detected in the active samples were not detected in the killed controls presumably due to the lack of biological activity. Of the 72 metabolites that were detected in the killed controls, all but 19 were significantly different ($p < 0.05$) from the active samples and qualitatively all show different dynamics (Supplementary Tables 1B and 2; Supplementary Figure 3).”

2. It's still not clear to me why they use a single gene only for their phylogenetic analysis. They have binned genome sequences; they should be able to utilize multiple genes and see how congruent the phylogenies are. Using a single gene to show the level of similarity between their isolates and the sequences from the metagenome seems problematic to me--even if ANI data are included.

We agree with the reviewer. We now report on 16 ribosomal protein phylogenetic markers from each of the four metagenome assembled genomes or MAGs (noting that some lack particular markers). We include their percent identity for the marker proteins at the amino acid level to their corresponding isolate and their best taxonomic hit (provided in the new Supplementary Table 6). These genes were found to be 72-100% identical between the isolates and their related MAGs and most taxonomic hits were congruent with the *rpO* gene used for initial analyses.

For example, here is a section of supplementary table 6:

	Bacillus sp. 1 MAG			Bacillus sp. 2 MAG		
	orf-id	percent identity (amino acid) to D1B51 isolate	taxonomy of best hit in nr	orf-id	percent identity (amino acid) to L2B47 isolate	taxonomy of best hit in nr
rpO	45791285_905321_26	86.3	Bacteria; Firmicutes; Bacilli; Bacillales; Bacillaceae; Bacillus	40892447_826483_9	87.67	Bacteria; Firmicutes; Bacilli; Bacillales; Bacillaceae; Bacillus
rpR	45791285_905321_26	80	Bacteria; Firmicutes; Bacilli; Bacillales; Bacillaceae; Bacillus	40907227_826711_12	76.67	Bacteria; Firmicutes; Bacilli; Bacillales; Bacillaceae; Bacillus
rpF	37223525_767924_12	73.18	Bacteria; Firmicutes; Bacilli; Bacillales; Bacillaceae; Bacillus	38693271_792303_24	71.51	Bacteria; Firmicutes; Bacilli; Bacillales; Bacillaceae; Bacillus
rpsH	37223525_767924_13	87.12	Bacteria; Firmicutes; Bacilli; Bacillales; Bacillaceae; Bacillus	17555616_405144_4	84.09	Bacteria; Firmicutes; Bacilli; Bacillales; Bacillaceae; Bacillus
rpI	37223525_767924_15	87.15	Bacteria; Firmicutes; Bacilli; Bacillales; Bacillaceae; Bacillus	29914607_643178_7	89.39	Bacteria; Firmicutes; Bacilli; Bacillales; Bacillaceae; Bacillus
rpIX	29907538_643074_6	81.55	Bacteria; Firmicutes; Bacilli; Bacillales; Bacillaceae; Bacillus	29914607_643178_6	85.44	Bacteria; Firmicutes; Bacilli; Bacillales; Bacillaceae; Bacillus
rpIN	40971199_827785_6	88.525	Bacteria; Firmicutes; Bacilli; Bacillales; Bacillaceae; Bacillus	29914607_643178_5	95.902	Bacteria; Firmicutes; Bacilli; Bacillales; Bacillaceae; Bacillus
rpsQ	40971199_827785_5	87.36	Bacteria; Firmicutes; Bacilli; Bacillales; Bacillaceae; Bacillus	29914607_643178_4	93.1	Bacteria; Firmicutes; Bacilli; Bacillales; Bacillaceae; Bacillus
rpP	40971199_827785_3	93.01	Bacteria; Firmicutes; Bacilli; Bacillales; Bacillaceae; Bacillus	29914607_643178_2	98.61	Bacteria; Firmicutes; Bacilli; Bacillales; Bacillaceae; Bacillus
rpsC	40971199_827785_2	86.301	Bacteria; Firmicutes; Bacilli; Bacillales; Bacillaceae; Bacillus	27147229_592856_6	88.94	Bacteria; Firmicutes; Bacilli; Bacillales; Bacillaceae; Bacillus
rpIV	40971199_827785_1	89.52	Bacteria; Firmicutes; Bacilli; Bacillales; Bacillaceae; Bacillus	1844421_47423_38	83.93	Bacteria; Firmicutes; Bacilli; Bacillales; Bacillaceae; Bacillus
rpsS	27828180_605619_3	85.87	Bacteria; Firmicutes; Bacilli; Bacillales; Bacillaceae; Bacillus	1844421_47423_37	91.3	Bacteria; Firmicutes; Bacilli; Bacillales; Bacillaceae; Bacillus
rpIB	27828180_605619_4	88.41	Bacteria; Firmicutes; Bacilli; Bacillales; Bacillaceae; Bacillus	1844421_47423_36	94.18	Bacteria; Firmicutes; Bacilli; Bacillales; Bacillaceae; Bacillus
rpID	27828180_605619_6	83.09	Bacteria; Firmicutes; Bacilli; Bacillales; Bacillaceae; Bacillus	1844421_47423_34	83.57	Bacteria; Firmicutes; Bacilli; Bacillales; Bacillaceae; Bacillus
rpIC	27828180_605619_7	82.13	Bacteria; Firmicutes; Bacilli; Bacillales; Bacillaceae; Bacillus	1844421_47423_33	89	Bacteria; Firmicutes; Bacilli; Bacillales; Bacillaceae; Bacillus
rpsJ	27828180_605619_8	92.16	Bacteria; Firmicutes; Bacilli; Bacillales; Bacillaceae; Bacillus	38693271_792303_40	97.06	Bacteria; Firmicutes; Bacilli; Bacillales; Bacillaceae; Bacillus

3. While they have done a good job in the discussion placing caveats around their interpretation of the exometabolites coming from or being consumed by their specific isolates, their results and Figure 3 are not nearly so careful in terms of language or clarification regarding their confidence that these isolates are truly representative of their nearest neighbours in the crusts. I still think that they need to tone down the language a bit, and to clarify that figure 3 is a

conceptual model and that those organisms labeled on the figure may be representative of metabolic groups.

We agree that the results section referring to figure 3 may have been misleading and we introduced the caveat that these bacteria may represent metabolically similar organisms (Line 239).

“We next combined our microbe-metabolite correlation data with the isolate exometabolomics data in order to visualize a wetting-induced dynamic exometabolomic foodweb that may result from the release of metabolites by a primary producer (e.g. Microcoleus sp.) and consumption of metabolites by heterotrophs (e.g. Bacillus sp. 1 and 2) (Figure 3). For simplicity, Blastococcus sp. is not shown in Figure 3 since this microorganism did not display sequential responses to wetting. It should be noted that the bacteria specified here likely represent metabolically similar groups of organisms that release/consume the same metabolites.”

And this is also now clarified in the discussion (Line 449):

“While this network portrays the dynamic substrate preferences of three specific native biocrust bacteria, there are many functionally-similar microorganisms that could fall into the categories of ‘early responders’, ‘mid-responders’ and ‘late-responders’ (grouping with Microcoleus sp. and the two Bacilli sp.) for a wetting event.”

Furthermore, the caption for Figure 3 has been modified (line 741):

“This network displays the relationships between metabolites and three dominant bacteria (or metabolically similar organisms) as they increase and decrease in relative abundance across wetting and successional stages in biocrust. The lower line plot corresponds to real relative abundance measurements for the three bacteria in level C successional stage biocrust. As Microcoleus sp. increases in relative abundance immediately after wetting, many metabolites released by the closest-related isolate are positively correlated with Microcoleus sp. in biocrust (solid red arrows) and as the two Bacilli increase in relative abundance (first Bacillus sp. 1 then Bacillus sp. 2), most metabolites consumed by the closest-related isolate decrease and are negatively correlated with these bacteria in biocrust (solid blue arrows) and most released metabolites are positively correlated (solid red arrows). Dotted arrows indicate metabolites that are released (red) or consumed (blue) by isolates, but did not display the expected relationship with that microorganism in situ. The thickness of the line corresponds to the absolute value of the Spearman’s rho correlation coefficient.”

Overall, my impression is that the manuscript is greatly improved but that it still suffers from a bit of overinterpretation of some data and incomplete analysis of other data.

Reviewer #4 (Remarks to the Author):

We thank the reviewer for their great attention to detail and very important suggestions. In response to these concerns, we have updated the link to the sequencing data, did a more thorough gene-gene comparison between isolate genomes and environmental metagenomes

using single-copy phylogenetic markers and further explained our metabolomics analysis so that it can be repeated by open-source software. Details can be found below.

Here are my comments to the reply of the authors:

'These data are now publicly available in NCBI SRA as indicated in the "data availability" section of the manuscript (line 539):

"Sequencing data that support the findings of this study are available at the NCBI Sequence Read Archive (accession number PRJNA395099, <https://www.ncbi.nlm.nih.gov/bioproject/?term=PRJNA395099>) and metabolomics data have been deposited to the EMBL-EBI MetaboLights database⁷¹ with the identifier MTBLS492 (<http://www.ebi.ac.uk/metabolights/MTBLS492>)."

The data is now at the NCBI. Just check one of the biosamples which seems to be repeated and has no SRA data linked to it.

<https://www.ncbi.nlm.nih.gov/biosample/7370990>

We apologize for the confusion. All 20 samples in NCBI have linked SRA data. The biosample the reviewer indicates was a mistake. Unfortunately it cannot be deleted, so we have provided a new link with the correct SRA file numbers specified (line 682):

"Sequencing data that support the findings of this study are available at the NCBI Sequence Read Archive (project accession number PRJNA395099, sample accession numbers SRX3024638- SRX3024657, <https://www.ncbi.nlm.nih.gov/bioproject/395099>)"

'As additional support for our rplO-based results, we performed additional analysis to investigate the relatedness between our isolates and the in situ taxa using ANI and AAI calculations (Line 313):'

I do not agree with this analysis. The authors used the ANI (average nucleotide identity) and AAI (average amino acid identity) but their results don't make sense:

'Isolates and their closest matching environmental relatives ranged between 86-92% identical in their rplO sequence and 73-94% for ANI and 52-86% for AAI (Table 1). While there is a general lack of consensus of valid isolate-to-native population comparisons, our values indicate matching at approximately the genus or species level'

In the case of ANI, it make more sense to use it as a genus or species identification parameter only when you have long DNA fragments like contigs. I don't understand how you get a higher ANI value than the AAI, it should be the other way around.

We thank the reviewer for pointing this out, which is something we needed to clarify in the manuscript. Here, ANI was calculated based as "whole genome" average nucleotide identity and AAI as "gene" average amino acid identity (for coding loci). In a reference provided within the text (Rodriguez-R and Konstantinidis, 2014) they show that within the context of cultivation-free identification of bacterial species (within/between species/genus) AAI is in most cases is smaller than ANI (refer to Figure 3 within the reference which shows the comparison of AAI and ANI for complete bacterial genomes stratified by taxonomic names (species, genus, and division)). Since ANI is more widely used and to avoid confusion, these values were kept in the main text (Table 1) and AAI was moved to a supplementary table (5).

These few additional details are now included in the text (line 168):

“Further comparative analyses were conducted to calculate genome average nucleotide identity (ANI) between isolate genomes and their related metagenome-assembled genomes (MAGs) to which the *rplO* gene was co-binned, which validated that each isolate and their closest matching environmental relative were of the same genus or species following the conventions described in a previous report²⁸ (Table 1). Additional validation was obtained by comparing gene average amino acid identity (AAI) (Supplementary Table 5) and other ribosomal protein genes from within each MAG to the corresponding isolate (Supplementary Table 6).”

Also, the authors are missing the point. If they really want to obtain more information about the taxonomy, they should use all the information contained in the shotgun reads. As I suggested, they can use a k-mer spectra based analysis such as CLARK or Kraken that are really easy to use and can be run in less than an hour. Alternatively, if they want to use single copy markers, they can use ANPHORA2 or MOCAT or MetaPhlan. These programs use several single copy gene markers.

I don't see the point on doing an analysis with just one marker when you have much more information...

We agree with the reviewer. We now report on 16 ribosomal protein phylogenetic markers from each of the four metagenome assembled genomes or MAGs (noting that some lack particular markers). We include their percent identity for the marker proteins at the amino acid level to their corresponding isolate and their best taxonomic hit (provided in the new Supplementary Table 6). These genes were found to be 72-100% identical between the isolates and their related MAGs and most taxonomic hits were congruent with the *rplO* gene used for initial analyses.

For example, here is a section of supplementary table 6:

	Bacillus sp. 1 MAG			Bacillus sp. 2 MAG		
	orf-id	percent identity (amino acid) to D1851 isolate	taxonomy of best hit in nr	orf-id	percent identity (amino acid) to L2B47 isolate	taxonomy of best hit in nr
rplO	45791285_905321_23	86.3	Bacteria; Firmicutes; Bacilli; Bacillales; Bacillaceae; Bacillus	40892447_826483_9	87.67	Bacteria; Firmicutes; Bacilli; Bacillales; Bacillaceae; Bacillus
rplR	45791285_905321_26	80	Bacteria; Firmicutes; Bacilli; Bacillales; Bacillaceae; Bacillus	40907227_826711_12	76.67	Bacteria; Firmicutes; Bacilli; Bacillales; Bacillaceae; Bacillus
rplF	37223525_767924_12	73.18	Bacteria; Firmicutes; Bacilli; Bacillales; Bacillaceae; Bacillus	38693271_792303_24	71.51	Bacteria; Firmicutes; Bacilli; Bacillales; Bacillaceae; Bacillus
rpsH	37223525_767924_13	87.12	Bacteria; Firmicutes; Bacilli; Bacillales; Bacillaceae; Bacillus	17555616_405144_4	84.09	Bacteria; Firmicutes; Bacilli; Bacillales; Bacillaceae; Bacillus
rplE	37223525_767924_15	87.15	Bacteria; Firmicutes; Bacilli; Bacillales; Bacillaceae; Bacillus	29914607_643178_7	89.39	Bacteria; Firmicutes; Bacilli; Bacillales; Bacillaceae; Bacillus
rplX	29907538_643074_6	81.55	Bacteria; Firmicutes; Bacilli; Bacillales; Bacillaceae; Bacillus	29914607_643178_6	85.44	Bacteria; Firmicutes; Bacilli; Bacillales; Bacillaceae; Bacillus
rplN	40971199_827785_6	88.525	Bacteria; Firmicutes; Bacilli; Bacillales; Bacillaceae; Bacillus	29914607_643178_5	95.902	Bacteria; Firmicutes; Bacilli; Bacillales; Bacillaceae; Bacillus
rpsQ	40971199_827785_5	87.36	Bacteria; Firmicutes; Bacilli; Bacillales; Bacillaceae; Bacillus	29914607_643178_4	93.1	Bacteria; Firmicutes; Bacilli; Bacillales; Bacillaceae; Bacillus
rplP	40971199_827785_3	93.01	Bacteria; Firmicutes; Bacilli; Bacillales; Bacillaceae; Bacillus	29914607_643178_2	98.61	Bacteria; Firmicutes; Bacilli; Bacillales; Bacillaceae; Bacillus
rpsC	40971199_827785_2	86.301	Bacteria; Firmicutes; Bacilli; Bacillales; Bacillaceae; Bacillus	27147229_592856_6	88.94	Bacteria; Firmicutes; Bacilli; Bacillales; Bacillaceae; Bacillus
rplV	40971199_827785_1	89.52	Bacteria; Firmicutes; Bacilli; Bacillales; Bacillaceae; Bacillus	1844421_47423_38	83.93	Bacteria; Firmicutes; Bacilli; Bacillales; Bacillaceae; Bacillus
rpsS	27828180_605619_3	85.87	Bacteria; Firmicutes; Bacilli; Bacillales; Bacillaceae; Bacillus	1844421_47423_37	91.3	Bacteria; Firmicutes; Bacilli; Bacillales; Bacillaceae; Bacillus
rplB	27828180_605619_4	88.41	Bacteria; Firmicutes; Bacilli; Bacillales; Bacillaceae; Bacillus	1844421_47423_36	94.18	Bacteria; Firmicutes; Bacilli; Bacillales; Bacillaceae; Bacillus
rplD	27828180_605619_6	83.09	Bacteria; Firmicutes; Bacilli; Bacillales; Bacillaceae; Bacillus	1844421_47423_34	83.57	Bacteria; Firmicutes; Bacilli; Bacillales; Bacillaceae; Bacillus
rplC	27828180_605619_7	82.13	Bacteria; Firmicutes; Bacilli; Bacillales; Bacillaceae; Bacillus	1844421_47423_33	89	Bacteria; Firmicutes; Bacilli; Bacillales; Bacillaceae; Bacillus
rpsJ	27828180_605619_8	92.16	Bacteria; Firmicutes; Bacilli; Bacillales; Bacillaceae; Bacillus	38693271_792303_40	97.06	Bacteria; Firmicutes; Bacilli; Bacillales; Bacillaceae; Bacillus

'Scripts used to analyze metabolomics data (Metabolite Atlas) are public and the link is now provided under "Code Availability" in the methods section (line 547): "Metabolite atlas can be accessed at <https://github.com/biorack/metatlas>."

Several of the links (Installation, Examples and Information) inside the project web page are broken. Also, they completely neglected my comment about the restricted access to Metabolite Atlas. Even if now the software to analyze the LCMS data, there is no access to the database.

My impression is that this is a software that will work at the National Energy Research Scientific Computing Center (NERSC) but not somewhere else...

We appreciate the feedback on Metabolite Atlas from the reviewer. While we have made this software publicly available through Github, is not very refined. We hope that at some point we can develop it to support third party installations. However, using the raw data now available at Metabolights and information provided in Supplementary Table 1, readers can reproduce a similar extraction of peak areas using open-source software such as MZmine. To address this point, we have revised the methods and indicate more thoroughly how the analysis was done starting at line 586:

"Metabolomics data were analyzed using Metabolite Atlas⁶⁷ (<https://github.com/biorack/metatlas>) in conjunction with the Python programming language to construct extracted ion chromatograms corresponding to metabolites previously detected in biocrust or contained within our in-house standards library. Note that this same analysis could be done with other open-source software packages. Authentic standards were then used to validate assignments based on two orthogonal data (accurate mass less than 15 ppm, retention time within 1 min and/or MS/MS matching major fragments) relative to standards and/or the Metlin database^{68,69} and are provided in Supplementary Table 1A. Metabolites that did not match orthogonal measures were classified as putative and are indicated by parentheses in figures."

Reviewer #4 (Remarks to the Author):

After reviewing the latest version of the manuscripts, I have no more comments and will endorse this work for publication.